# Energy management strategy for methanol hybrid commercial vehicles based on improved dung beetle algorithm optimization

Zhihao Li[1,2], Ping Xiao[1,3]*, Jiabao Pan[1,2], Wenjun Pei[1,2], Aoning Lv[2]

1 Anhui Province Key Laboratory of Intelligent Car Wire-Controlled Chassis System, Anhui Polytechnic University, Wuhu, China, 2 School of Mechanical and Automotive Engineering, Anhui Polytechnic University, Wuhu, China, 3 School of Engineering, University of Brigeport, Brigeport, Connecticut, United States of America

* tlxp95@ahpu.edu.cn

**Data Availability Statement:** Data relevant to this study are available from the figshare database and

## Abstract

In order to solve the problem of poor adaptability and robustness of the rule-based energy management strategy (EMS) in hybrid commercial vehicles, leading to suboptimal vehicle economy, this paper proposes an improved dung beetle algorithm (DBO) optimized multi-fuzzy control EMS. First, the rule-based EMS is established by dividing the efficient working areas of the methanol engine and power battery. The Tent chaotic mapping is then used to integrate strategies of cosine, Lévy flight, and Cauchy Gaussian mutation, improving the DBO. This integration compensates for the traditional dung beetle algorithm's tendency to fall into local optima and enhances its global search capability. Subsequently, fuzzy controllers for the driving charging mode and hybrid driving mode are designed under this rule-based EMS. Finally, the improved DBO is used to obtain the optimal control of the fuzzy controller by taking the fuel consumption of the whole vehicle and the fluctuation change of the battery state of charge (*SOC*) as the optimization objectives. Compared to traditional rule-based energy management strategies, the optimized fuzzy control using the enhanced DBO continuously adjusts the torque distribution between the engine and motor based on the vehicle's real-time state, resulting in a 9.07% reduction in fuel consumption and a 3.43% decrease in battery *SOC* fluctuations.

## 1 Introduction

As energy shortage and environmental protection issues become increasingly pressing, the world's major automobile countries are actively carrying out the research and development of new energy vehicles. Hybrid electric vehicles (HEVs) can optimize energy management and thermal management systems to enhance heat transfer efficiency within the vehicle's energy transmission system. According to Animasaun et al. [1], the efficiency of heat transfer processes plays a crucial role in addressing energy shortages. Efficient heat transfer determines the effectiveness of energy utilization and conservation within a system, directly influencing overall energy demand and supply dynamics. Commercial vehicles are relatively lagging behind in

the link is:https://doi.org/10.6084/m9.figshare.26229455.v1.

**Funding:** The research was supported by Key Research and Development Projects in Anhui Province (2022a05020007) and the National Natural Science Foundation of China (52375227). Professors Ping Xiao and Jiabao Pan made substantial contributions to the research direction design, data analysis, and manuscript revisions.

**Competing interests:** The author(s) declared no potential conflicts of interest with respect to the research, authorship, and/or publication of this article.

their new energy transformation due to factors such as large load capacity, complex working environment, high cost, and regulations and policies [2].Compared with pure electric and fuel cell commercial vehicles, methanol hybrid commercial vehicles are less restricted by battery technology. Methanol, as a vehicle fuel alternative, holds promise for easing petroleum shortages and further improving economic and environmental sustainability [3]. With relatively low transition costs and easy industrialization, methanol hybrid commercial vehicles are the trend for future development of hybrid commercial vehicle.

The EMS of hybrid vehicles is the core technology, which allocates the torque distribution between the engine and the electric motor in real time during the driving process of the whole vehicle, and the goodness of the EMS affects the economy of the whole vehicle to a large extent. The EMS of hybrid vehicles is the core technology, and the goodness of EMS greatly affects the economy of the whole vehicle. The EMS of hybrid vehicles are mainly divided into four types: energy management strategies based on deterministic rules, energy management strategies based on fuzzy rules, energy management strategies based on global optimization and energy management strategies based on instantaneous optimization [4–6].

The rule-based deterministic EMS assigns fixed control parameters and switches the vehicle to different modes based on its current state, adhering to predetermined limits. This approach primarily relies on subjective experience to optimize economic efficiency. Davis et al. [7] proposes a rule-based adaptive strategy to realize the adjustment of battery charging power according to the change of vehicle demand power. Jeoung et al. [8] obtains the optimal logic threshold value through comparative optimization to achieve the best economic efficiency. The use of swarm intelligence optimization algorithms to find the optimal logic thresholds with the objective of economy and emission minimization[9–11], which significantly improves fuel economy is gaining more and more English in energy management strategies. Although this strategy is simple and efficient, it has poor adaptability and robustness, and cannot be adjusted and optimized with the real-time state of the vehicle.

The fuzzy rule-based EMS fuzzyfies the precise values of inputs and uses fuzzy control rules to make decisions on the operating state of the vehicle to achieve regulation and control of the system, thereby improving the adaptability and robustness of energy management. Li et al. [12] proposes a fuzzy EMS in which the real-time state of the vehicle is used as an input, and the vehicle is able to adjust the torque distribution with the real-time state changes, enhancing the flexibility and adaptability of control. Lei et al. [13] proposes the use of a dual-mode controller to control the drive and brake energy recovery, which improves the fuel economy by 6.7% compared to the conventional single fuzzy controller. However, the above study did not consider the driving conditions in the driving charging mode and hybrid driving mode, the two battery *SOC* working area is not the same, and the setting of fuzzy control affiliation and rules are mostly affected by subjective factors, and can not achieve the global optimization.

The EMS based on global optimization is based on a cost function that has been defined and a minimum cost function for the driving conditions, resulting in the best fuel economy for the vehicle. The control parameters are selected through dynamic planning and applied to a rule-based EMS to significantly improve the overall vehicle economy [14, 15]. Mashadi et al. [16] utilized dynamic programming in combination with particle swarm algorithm to determine the optimal operating conditions of the powertrain and then develop control rules. Anselma et al. [17] added the slope weighting method in dynamic programming to improve the computational speed of dynamic programming, which not only greatly improves the computational speed, but also limits its fuel increment relative to the traditional dynamic programming to less than 3.3%. However, the dynamic planning algorithm is complicated to

calculate and requires the data of the whole working condition, which makes the practical application relatively difficult.

Energy management strategies based on transient optimization are divided into equivalent fuel minimum energy management strategies and model predictive energy management strategies. Transient optimization involves selecting the equivalent fuel consumption or total power consumption as the objective function for optimization at each moment of vehicle operation, thus determining the instantaneous optimal operating point for distributing torque between the engine and the electric motor. Deng et al. [18] used a dynamic programming algorithm to obtain the global optimum equivalent factor, and applied the equivalent factor to the constructed equivalent fuel-minimum control model. Wang et al. [19] used a fuzzy control algorithm as well as an optimization algorithm to find the optimal equivalent factor, respectively. However, the process is complicated and depends on accurate models and data, which is difficult to realize. Xue et al. [20] Model prediction combined with an adaptive equivalent fuel consumption minimization strategy leads to power allocation at planned speeds, thus ensuring fuel economy, adaptability, and global optimality. Zhou et al. [21] designed a mixed-model predictive controller for optimizing a multi-objective EMS problem under vehicle following state to achieve fuel efficiency improvement. But the model prediction needs to solve the optimization problem all the time, which leads to a very complicated computation, and the equivalent fuel minimization strategy is highly dependent on the equivalence factor and the dynamic driving situation, which makes it relatively difficult to be applied in practice.

In view of the above problems, this paper proposes an improved DBO optimized multi-fuzzy control EMS for hybrid commercial vehicles. Unlike conventional fuel-hybrid commercial vehicles, this paper utilizes a more environmentally friendly and economical methanol engine. Additionally, compared to the traditional fuzzy control EMS, we incorporate fuzzy control to optimize both the driving charging mode and the hybrid drive mode. This approach avoids the problem of excess engine output capacity caused by different battery *SOC* working areas in the two modes. In order to achieve the most economical control strategy for the entire vehicle, this paper employs the DBO to optimize the fuzzy controller. However, to address the DBO's shortcomings, such as susceptibility to local optima and slow convergence, we integrate several enhancements: Tent chaotic mapping for population initialization with positive cosine random assignment, a dung beetle forager fused with the Lévy flight strategy, and the Cauchy Gaussian mutation strategy. These improvements enhance the global optimization speed, convergence speed, and search accuracy of the DBO. The improved DBO is then used to optimize the fuzzy control with the objectives of maximizing vehicle economy and minimizing battery *SOC* fluctuations. The results indicate that the improved DBO exhibits faster convergence speed, higher convergence accuracy, and greater stability compared to the traditional DBO. This leads to improved vehicle economy and reduced battery *SOC* fluctuations when optimizing the EMS with multi-fuzzy control.

The structure of this paper is as follows: First, a vehicle simulation model is established in AVL Cruise. Second, a rule-based EMS is developed in Simulink. Subsequently, an improved DBO is introduced and compared with the original DBO, genetic algorithm (GA), and particle swarm algorithm (PSO). By incorporating fuzzy control in the driving charging mode and hybrid drive mode, a multi-fuzzy control EMS is established. Then, the improved DBO is used to optimize the fuzzy control, resulting in an EMS based on the optimized fuzzy control. Finally, the feasibility of the proposed method is verified and analyzed through simulation results. Flowchart of the article is shown in Fig 1.

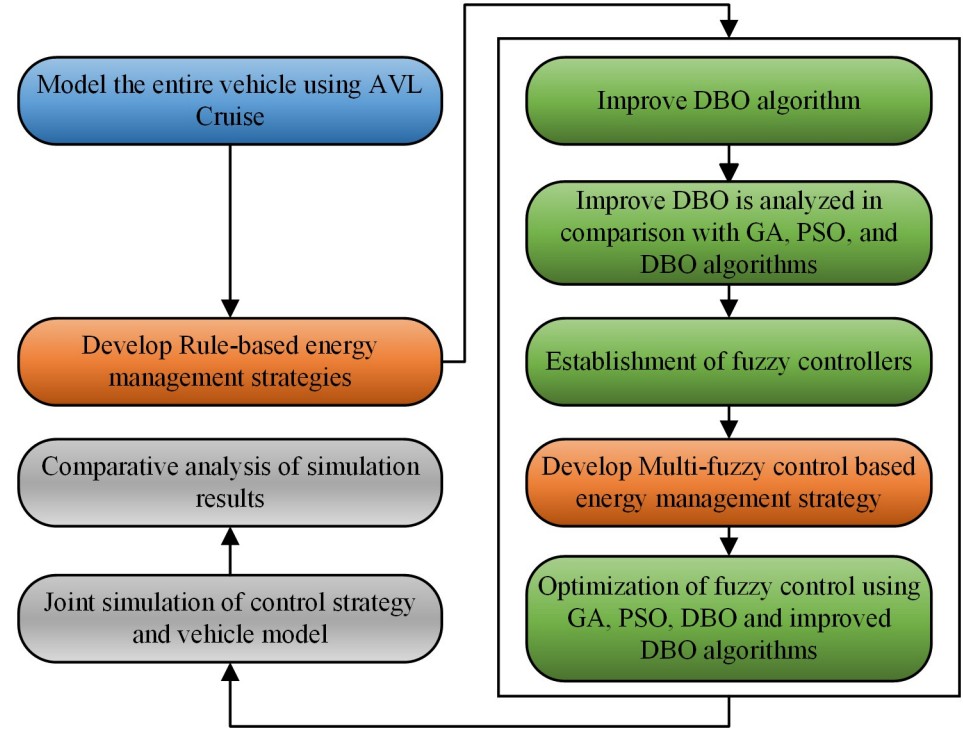

**Fig 1. Article flowchart.**

## 2 Modeling and simulation of parallel hybrid heavy duty commercial vehicle

### 2.1 Parallel hybrid heavy commercial vehicle architecture

This paper studies a heavy-duty hybrid commercial vehicle with a methanol engine, and the structure of the system is shown in Fig 2. The whole vehicle has torque inputs from two power sources, the engine and the electric motor, and the engine is connected to the electric motor through a clutch, which is opened and closed by controlling the clutch so as to realize the

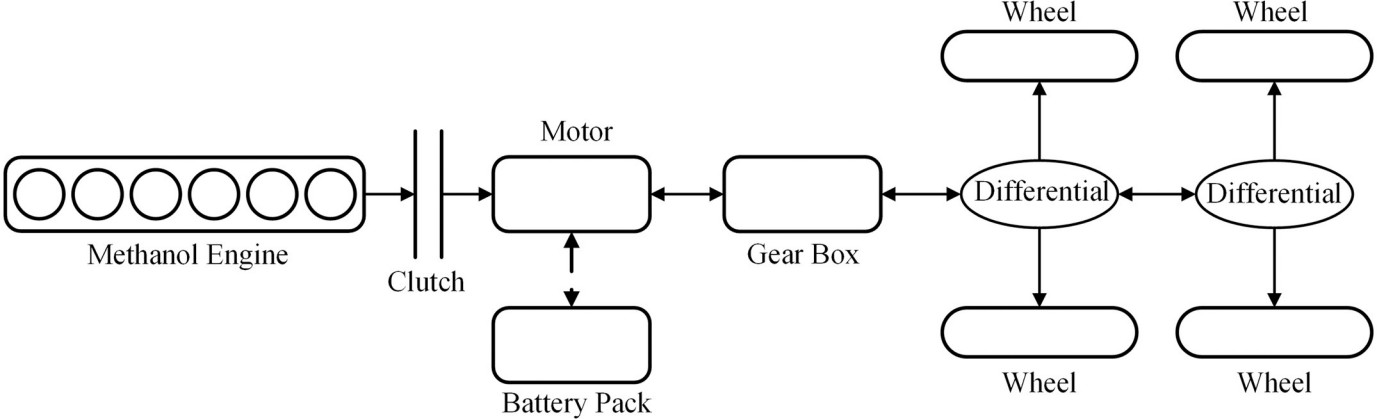

**Fig 2. Hybrid heavy commercial vehicle structure schematic diagram.**

**Table 1. Vehicle parameters.**

| Parameters | Value |
|---|---|
| Overall mass m/kg | 8700 |
| Full load mass $m_1$/kg | 49000 |
| Wheel Rolling Radius r/m | 0.521 |
| Wind resistance $C_D$ | 0.5 |
| Windward side A/m$^2$ | 8.8 |
| Single ratio transmission | 2.533 |

switching of driving modes. Hybrid electric vehicles can recharge the power battery when the battery *SOC* falls below a set value, maintaining the balance of the battery *SOC*. This can reduce the dependence of hybrid vehicles on charging stations during operation and minimize the negative impact on the microgrid [22]. The main parameters of the whole vehicle are shown in Table 1:

## 2.2 Vehicle dynamics model

According to the analysis of the longitudinal forces during the travel of the vehicle and based on the resistance that the car must overcome on the road, the equation of travel of the vehicle is obtained as:

$$F_t = F_f + F_w + F_i + F_j \tag{1}$$

Where $F_t$ is the driving force; $F_f$ is the rolling resistance; $F_w$ is the air resistance: $F_i$ is the slope resistance; $F_j$ is the acceleration resistance.

$$F_f = fmg \tag{2}$$

Where $f$ is the rolling resistance coefficient; $m$ is the vehicle mass. The rolling resistance coefficient is affected by the road surface condition and vehicle speed.

$$F_w = \frac{C_D A v_a^2}{21.15} \tag{3}$$

Where $C_D$ is the air resistance coefficient; $A$ is the windward area of the vehicle; $v_a$ is the traveling speed of the vehicle.

$$F_i = G \sin \alpha \tag{4}$$

Where $\alpha$ is the inclination angle of the ramp; $G$ is the vehicle gravity.

$$F_j = \delta m \frac{dv_a}{dt} \tag{5}$$

Where $\delta$ is the mass proportionality coefficient; the driving force required to move the vehicle, $F_t$ is shown in Eq (6):

$$F_t = mgf + \frac{C_D A v_a}{21.15} + G \sin \alpha + \delta m \frac{dv_a}{dt} \tag{6}$$

Where $i_g$ is the transmission ratio; $i_0$ is the main reduction ratio; $\eta$ is the mechanical efficiency of the entire driveline; $r$ is the wheel rolling radius; that is the demand torque $T_{req}$ is shown in

Eq (7):

$$T_{req} = \frac{F_t r}{i_g i_0 \eta} \tag{7}$$

## 2.3 Methanol engine model

Methanol, which has a higher content of oxygen atoms, is more easily and completely combusted, resulting in lower emissions [23]. The development of alternative fuels for automobiles is one of the most effective ways to solve the shortage of oil resources, and methanol is an alternative fuel for automobiles with great potential for application and market prospects. On the one hand, compared with other alternative fuels, the industrial chain of methanol production and manufacturing is mature, with a good industrial base and a large industrial scale, and it is liquid at room temperature and pressure, which makes it easy to be transported and used. The engine used in this paper is 15L methanol engine. The parameters are shown in Table 2.

Since the focus of this paper is not on the dynamic characteristics of the engine, the engine is simplified to a static model for the convenience of the study, and its engine universal diagram is shown in Fig 3. The transient engine methanol consumption rate can be expressed by Eq (8), which is a function of engine speed and torque.

$$\dot{m}_f = f(T_e(t), n_e(t)) \tag{8}$$

Where $\dot{m}_f$ is the engine methanol consumption rate; $T_e(t)$ is the engine torque; $n_e(t)$ is the engine speed.

## 2.4 Motor model

The selection of the electric motor is critical to the overall EMS, and the motor should be able to replace the engine in its inefficient operating zone during the entire vehicle's travel, ensuring that the engine operates in its efficient zone. Table 3 shows the parameters of the motor.

The motor map is obtained from the motor speed, torque, and efficiency as shown in Fig 4. The power of the motor can be expressed as a function of motor torque, speed and efficiency. Eqs (9) and (10) indicate that the motor is used as a drive motor as well as a generator, respectively.

$$p_{em}(t) = \frac{T_{em}(t) \cdot n_{em}(t)}{9550 \eta_m(t)} \tag{9}$$

$$p_{em}(t) = \frac{T_{em}(t) \cdot n_{em}(t) \cdot \eta_g(t)}{9550} \tag{10}$$

Where $p_{em}(t)$ is the motor power; $T_{em}(t)$ is the motor torque; $n_{em}(t)$ is the motor speed; $\eta_m$

**Table 2. Methanol engine parameters.**

| Parameters | Value |
|---|---|
| Engine displacement (L) | 15 |
| Peak torque (N.m) | 2600 |
| Peak power (Kw) | 380 |
| Peak speed (r/min) | 2100 |

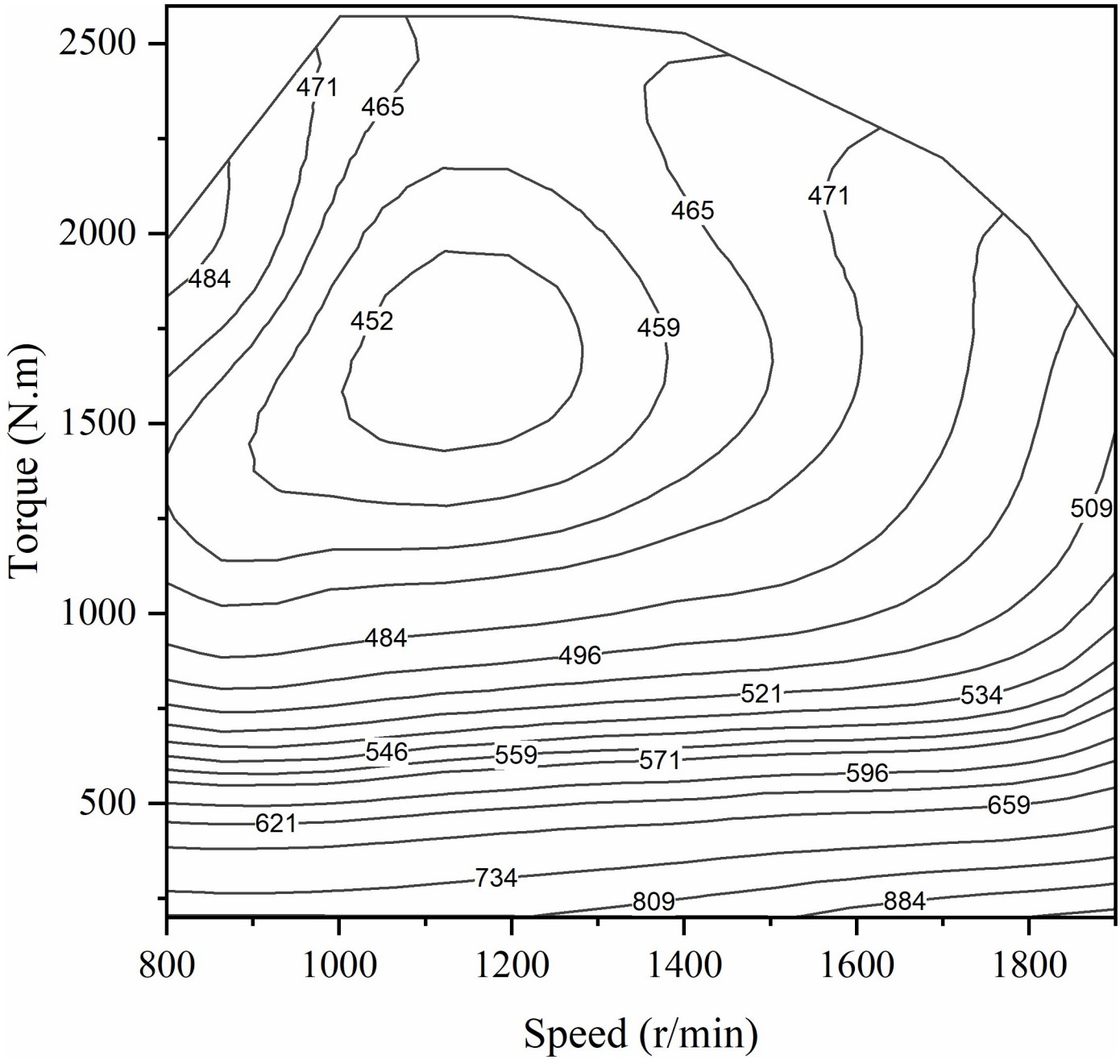

**Fig 3. Engine methanol consumption contour plot.**

$(t)$ is the efficiency when the motor is driven; $\eta_g(t)$ is the efficiency of the motor when generating electricity.

## 2.5 Vehicle simulation modeling

In order to obtain simulation results close to the real driving conditions, this paper utilizes AVL Cruise software to establish the physical model, establishes the control strategy in

**Table 3. Motor parameters.**

| Parameters | Value |
|---|---|
| Peak power (Kw) | 180 |
| Rated power (Kw) | 120 |
| Peak torque (N. m) | 1560 |
| Rated torque (N. m) | 1000 |
| Peak speed (r/min) | 2360 |

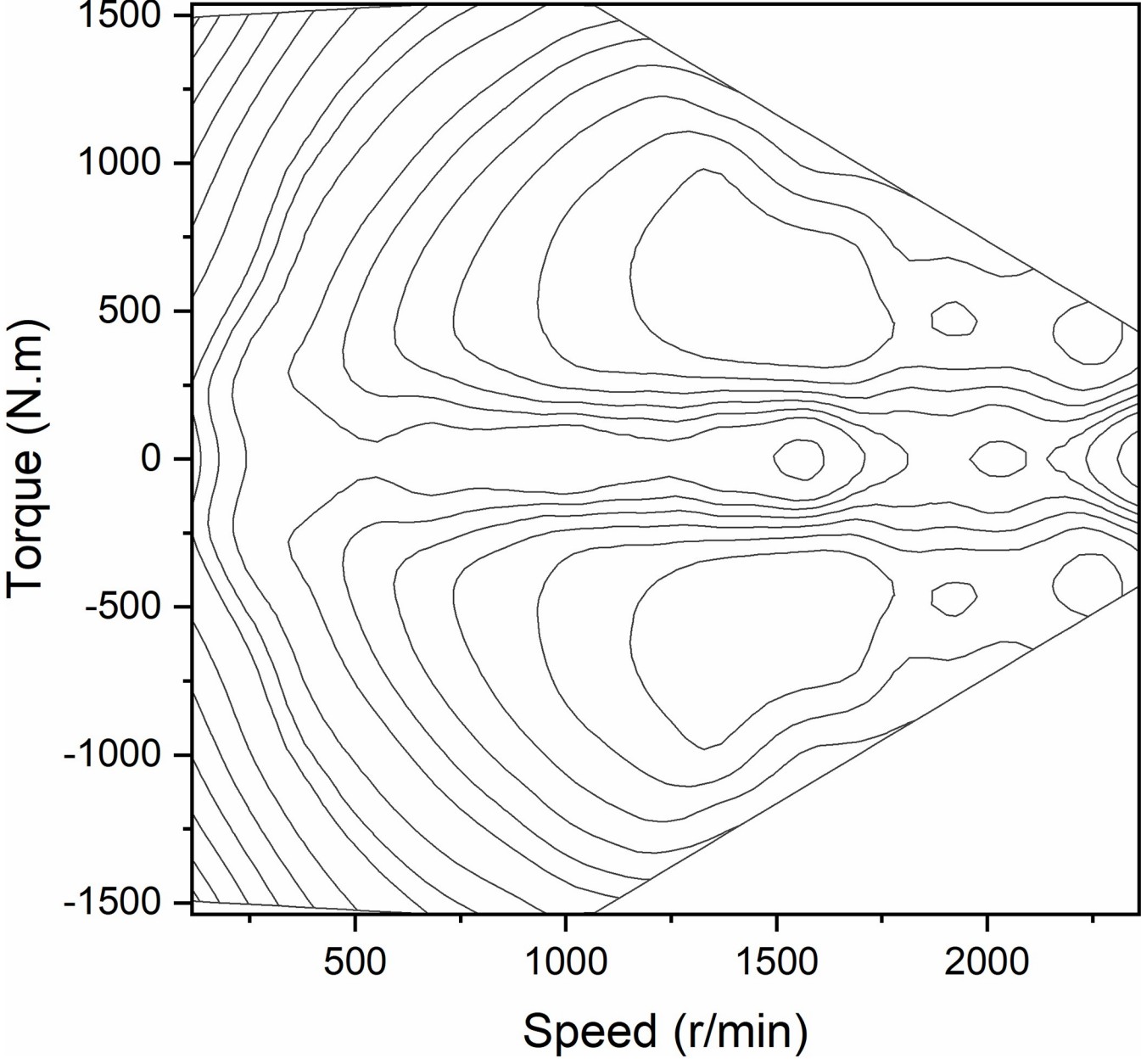

**Fig 4. Motor efficiency map.**

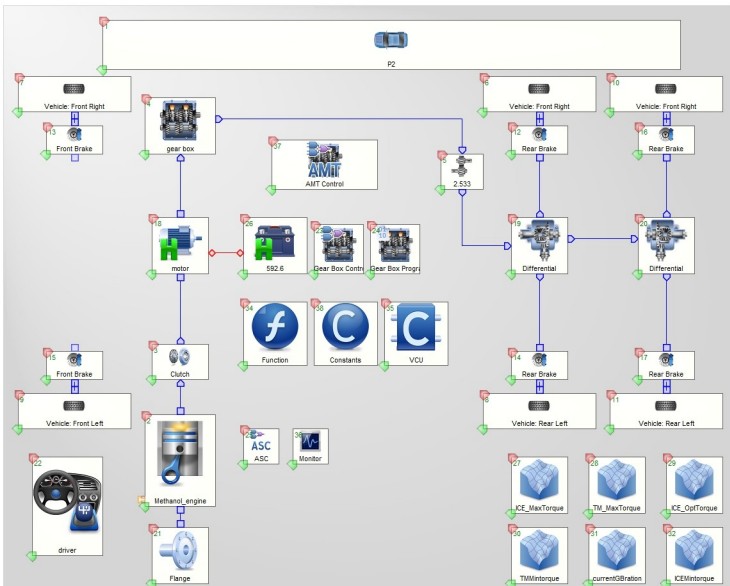

**Fig 5. Physical modeling of parallel hybrid electric vehicles.**

Matlab/Simulink software, and utilizes the co-simulation interface to make the established control strategy as the vehicle control unit (VCU). As shown in Fig 5.

## 3 Rule-based control strategy working mode switching and working principle

The key to the design of the rule-based torque coordination control strategy is to utilize the logic threshold value parameter, the optimal operating region of the engine and the power battery to control the switching of the operation mode and the torque allocation. The logic threshold value parameters are set as in Table 4.

The EMS for hybrid vehicles aims to keep the engine operating in its high-efficiency zone while limiting the $SOC$ variations of the power battery, avoiding overcharging and over-discharging to extend battery life. In order to realize the above purpose, the efficient operating zone of the engine needs to be determined first. Its high-efficiency speed zone is set as

**Table 4. Logic threshold settings.**

| Parameters | Parameter Description | Unit |
|---|---|---|
| $T_{req}$ | Vehicle Demand Torque | $Nm$ |
| $T_{e\_max}$ | Maximum engine output torque | $Nm$ |
| $T_{e\_opt}$ | Optimum engine output torque | $Nm$ |
| $T_{e\_min}$ | Engine Limit Output Torque | $Nm$ |
| $T_{m\_max}$ | Maximum motor output torque | $Nm$ |
| $V_{max}$ | Maximum speed for pure electric vehicles | $km$ |
| BSP | Brake Pedal | / |
| $SOC_{max}$ | $SOC$ Maximum | % |
| $SOC_{targrt}$ | $SOC$ target value | % |
| $SOC_{min}$ | $SOC$ Min | % |

1000rpm-1400rpm,and the maximum output curve $T_{max}$, the optimal economic curve $T_{Opt}$, and the minimum torque output $T_{min}$, and the high-efficiency working zone of the engine in this paper is shown in Fig 6.

According to the power battery's charging and discharging internal resistance change curve, delineate the minimum $SOC_{min}$ and maximum $SOC_{max}$ value of the battery $SOC$ work, so that the battery is always working in the high efficiency zone, but in the actual operation process we need to set a target value of $SOC$, as shown in Fig 7. When the battery is lower than

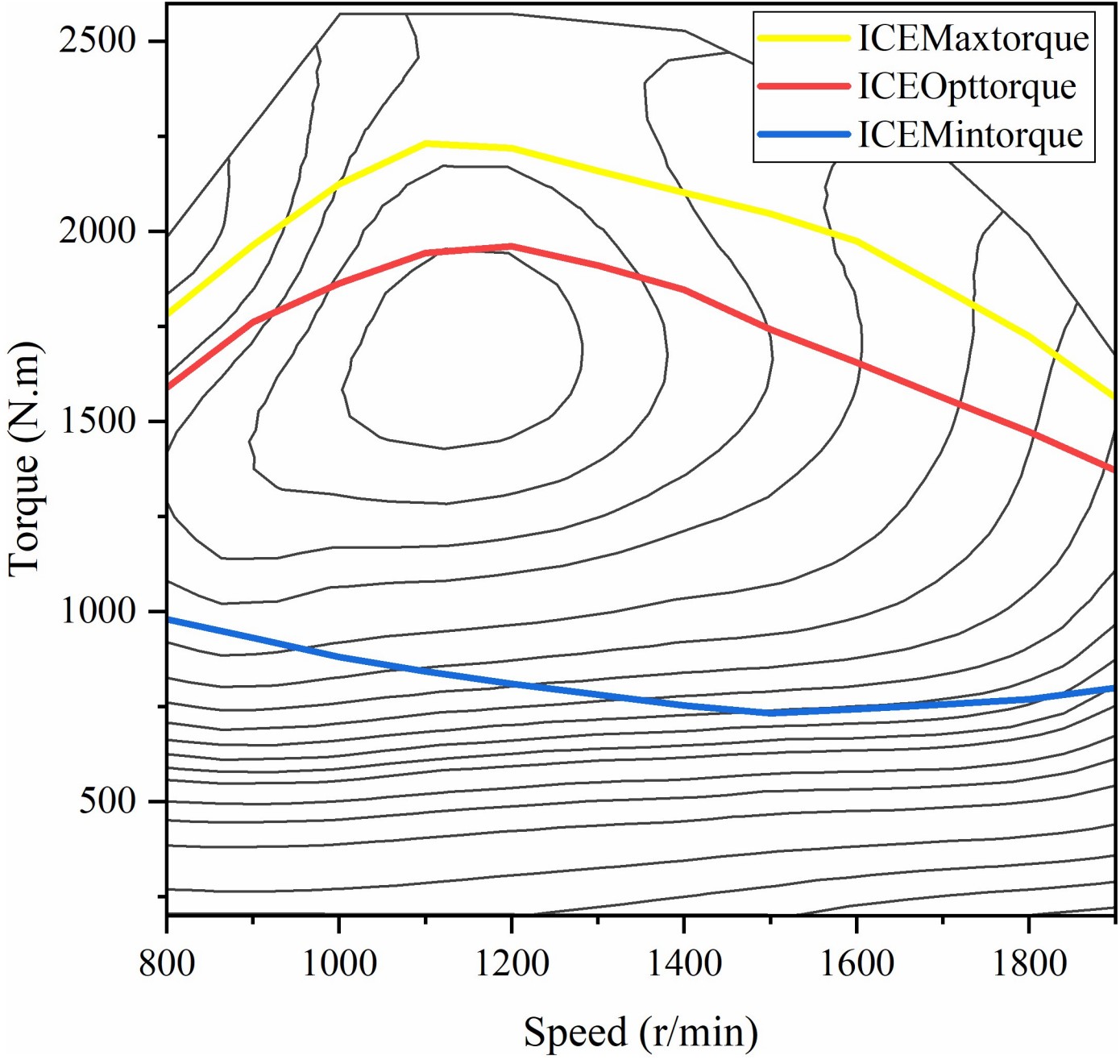

**Fig 6. Division of engine working area.**

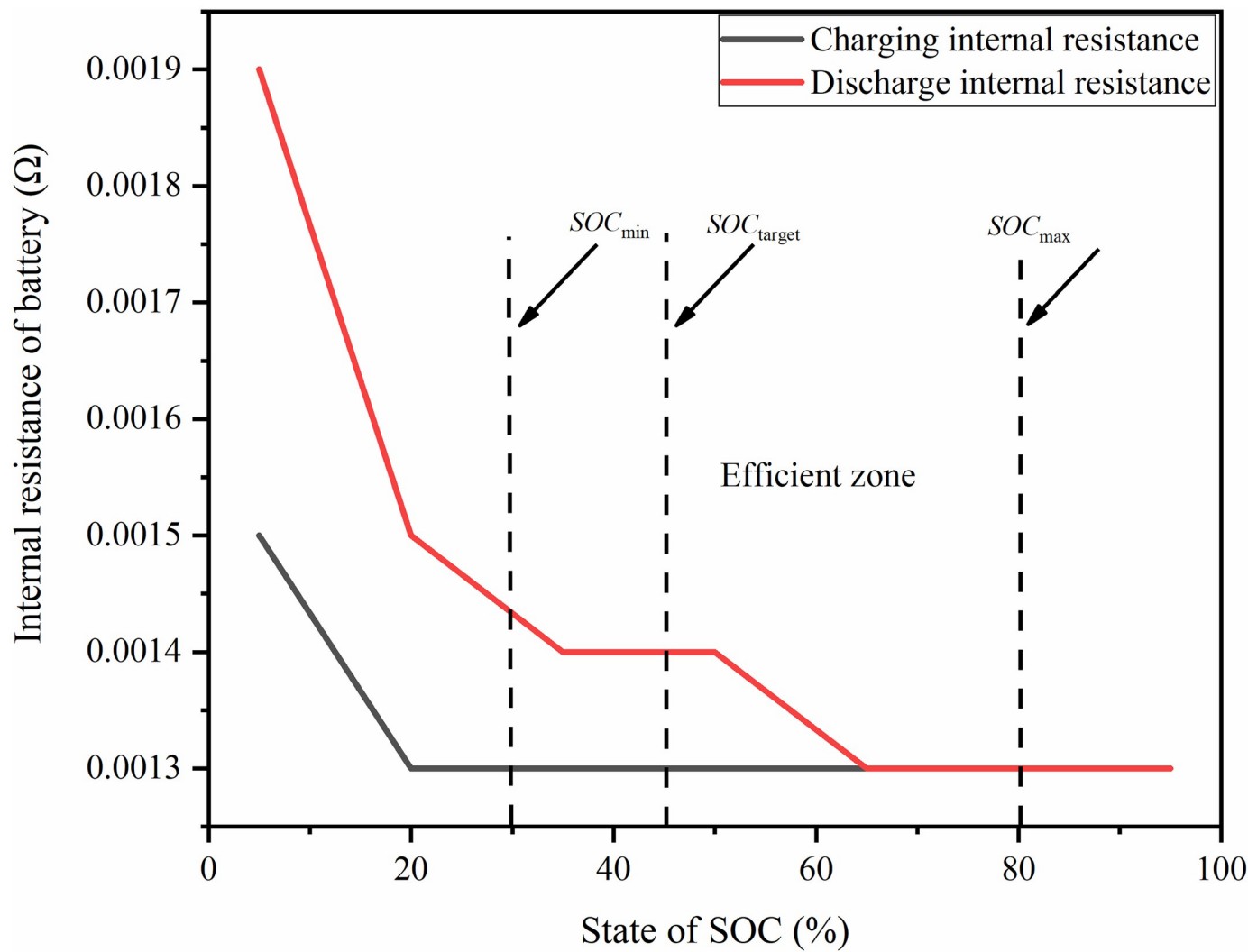

**Fig 7. Division of power battery working area.**

this target value for replenishment, this paper sets the battery to work in the range of [0.3 0.8], and sets the target value of the battery *SOC* to 0.45, so as to ensure that the power is maintained during the driving process, and to avoid excessive charging and discharging of the battery.

Since the power source of the whole vehicle consists of the engine and the motor together, the optimal torque distribution is realized in order to improve the economy of the whole vehicle. The operation modes and switching conditions are shown in Table 5.

## 4 Fuzzy control energy management strategy based on improved dung beetle algorithm optimization

In the rule-based control strategy, the allocation of torque can only be deployed according to human regulations, and the adaptability is poor, so most scholars integrate fuzzy controllers into control strategy to improve the robustness and adaptability of the control strategy. In this paper, the improved DBO is used to optimize the multi-fuzzy controller to maximize the

**Table 5. Operating modes and switching conditions.**

| Current Mode | Switching Condition | Target Model |
|---|---|---|
| Park mode | $T_{req} \leq T_{m\_max}$ & $SOC_{min} \leq SOC$; $V \leq V_{max}$; | all-electric |
| all-electric | $T_{req} \leq T_{e\_opt}$ & $SOC_{target} \geq SOC$; | Vehicle Charging |
| Vehicle Charging | $T_{m\_max} \leq T_{req} \leq T_{e\_max}$ & $n_e \geq 1000$; $SOC_{min} \geq SOC$; | Direct Engine Drive |
| Direct Engine Drive | $T_{req} \geq T_{e\_max}$ & $SOC_{min} \leq SOC$; | Hybrid Drive |
| Hybrid Drive | $BSP > 0$ & $SOC_{max} \geq SOC$; | Energy Recovery |
| Energy Recovery | $T_{req} = 0$; | Park mode |

economic efficiency of the whole vehicle under the premise of ensuring the dynamics of the whole vehicle.

## 4.1 Improving the dung beetle algorithm

Despite its outstanding performance, the DBO algorithm still has some problems that need to be solved, such as the relative lack of global exploration capability, which makes it easy to fall into local optimal solutions [24]. For applications that require accurate optimization in the problem solution space, the algorithm must have both good global search capability and local search capability. To address this, the Tent chaotic mapping fusion positive cosine population initialization improvement strategy, the dung beetle forager fusion Lévy flight strategy, and the Cauchy Gaussian variation strategy are incorporated into the DBO, and the Chaotic Cauchy Variation DBO (TLK-DBO) is developed.

(1) Tent chaotic mapping combined with sine and cosine population initialization improvement strategy. Population initialization significantly influences the convergence speed and optimal solutions of global optimization searches. The traditional random distribution used in DBO algorithm population initialization may fail to ensure diversity and effectiveness, whereas Tent chaotic mapping is renowned for its extensive randomness, continuity, smoothness, and enhanced diversity. These properties together enable the initial population to uniformly cover the entire search space, thereby avoiding premature convergence to local optima and ensuring a smooth search process that balances exploration and exploitation needs, thereby enhancing the algorithm's global search capability and robustness [25].Additionally, this paper integrates Tent chaotic mapping with sine and cosine random assignment strategies, leveraging the advantages of chaotic and trigonometric mappings to further optimize population initialization. The sine and cosine strategies contribute to the breadth and smoothness of the search process, ensuring not only diversity within the population but also uniform distribution across the entire search space. The mathematical model is as follows:

$$x_{i+1} = \begin{cases} 2 \cdot x_i & x_i < 0.5 + \epsilon \\ 2 \cdot (1 - x_i) & x_i \geq 0.5 + \epsilon \end{cases} \tag{11}$$

$$x_{i+1} = x_{i+1} + \alpha \cdot \sin(2\pi x_{i+1}) + \alpha\cos(2\pi x_{i+1}) \tag{12}$$

The use of Tent chaotic mapping when the value is equal to 0.5 will cause the initialization to fall into a short cycle, in order to avoid this, so that a very small value is added to ensure that it will not be equal to 0.5. At the same time, in order to reduce the emergence of duplicate populations, the chaotic mapping process is coupled with a positive cosine random assignment strategy. Eq (12) is a random assignment value ranging from [0 0.2]. Fig 8 shows the

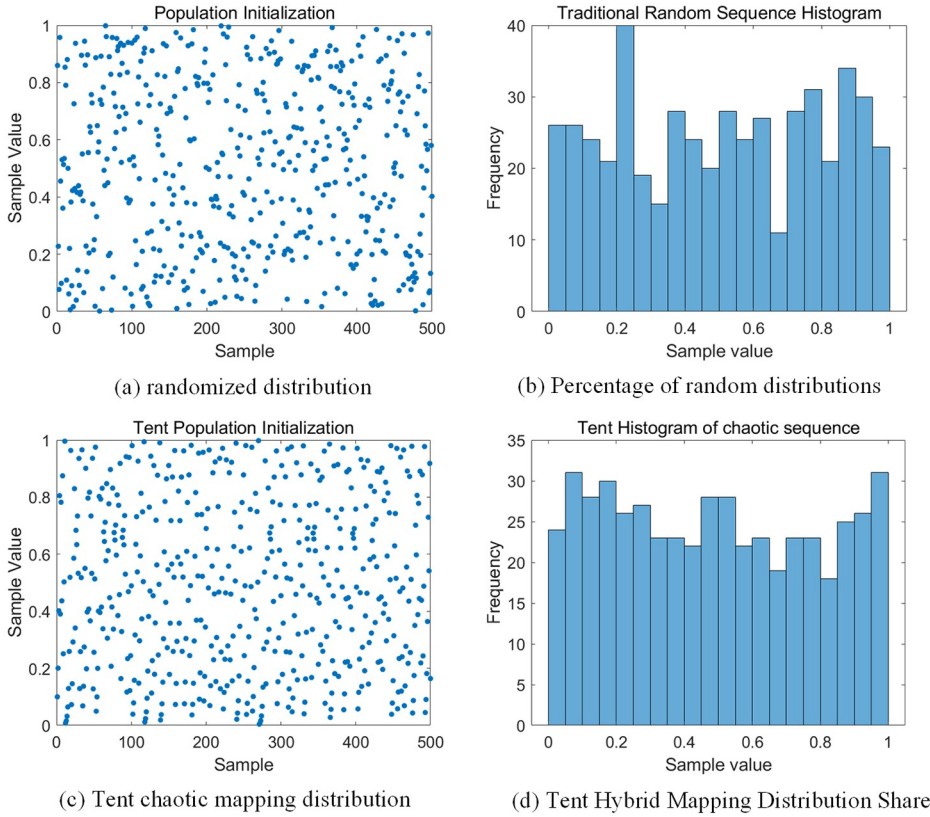

**Fig 8. Distribution of initial population.**

distribution and histogram of the initialization of the population of the DBO algorithm, and the comparison of the figure shows that the repetition of the initialization of the population in the TLK-DBO algorithm is significantly reduced, the distribution of the population is more uniform, and the randomness is higher.

(2) Dung beetle foragers incorporate Lévy flight strategies. Lévy flights utilize their long-tailed distribution to effectively prevent algorithms from falling into local optima, thereby enhancing their ability to discover global optimal solutions. They can generate large step sizes, accelerating the algorithm's exploration of the entire search space and improving the efficiency of finding optimal solutions to optimization problems. This strategy is suitable for various complex and diverse optimization scenarios, showing strong adaptability without being limited by specific problem types. Additionally, Lévy flights maintain stable search performance and excel in handling complex, high-dimensional, and nonlinear problems, thereby enhancing the algorithm's robustness and practicality [26]. The forager in the DBO will forage in the generated optimal foraging area, and with the addition of Lévy flight the forager is able to perform an all-around search in the optimal foraging area to improve the global search. Introducing the dung beetle forager position update, the forager position update formula is as follows:

$$x_i(t+1) = x_i(t) + Levy * (c_1 * (x_i(t) - Lb^b) + c_2 * (x_i(t) - Ub^b)) \tag{13}$$

(3) Cauchy Gaussian Variation Strategy. The DBO, like other swarm intelligence optimization algorithms, is prone to fall into local optimal solutions. When the optimal solution of the population after three consecutive iterations remains unchanged, the algorithm is judged to be

trapped in the local optimal solution, and at this time, the auxiliary strategy intervenes and uses the Cauchy Gaussian mutation strategy to make the algorithm jump out of the local optimal solution. The Cauchy Gaussian mutation strategy utilizes the randomness of the Cauchy distribution to effectively enhance the algorithm's global search capability and robustness. It allows the algorithm to jump to distant positions, avoiding the risk of premature convergence to local optimal solutions, thereby enabling greater exploration and variability. This strategy enhances the algorithm's exploratory and diverse capabilities in complex, high-dimensional, and nonlinear optimization problems, significantly increasing the probability of finding global optimal solutions. Moreover, it is versatile and practical, as it is not limited by specific problem types or parameter settings [27]. The Cauchy Gauss variation assisted jumping out of the local optimum method is given in the following Eq (14):

$$x_{newi}(t) = x_i(t) + \sigma_{cauchy}*\tan(\pi*(rand_i - 0.5)) + \sigma_{gaussian}*randn_i \tag{14}$$

Where $\sigma_{cauchy} = 1 - (t/m)^2$, is the scale parameter of the Cauchy distribution, $\sigma_{gaussian} = (t/m)^2$, is the standard deviation of the Gaussian distribution. The Cauchy variant takes a wider value to enhance the global search ability of the algorithm, and the Gaussian variant is more intensive to make up for the ability of local search, the Cauchy variant in the early stage can be a fast global search, and the Gaussian variant in the later stage enhances the local search ability of the algorithm, which can accelerate the algorithm iteration speed. Through improvement, the flow of the TLK-DBO algorithm is shown in Fig 9.

In order to verify the optimization ability of TLK-DBO algorithm, 8 test functions (4 single-peak benchmark functions and 4 multimodal benchmark functions) are selected, as shown in Table 6, to evaluate the global solution ability and the ability to jump out of the local optimum of the TLK-DBO algorithm by comparing with the GA, PSO, DBO algorithm. To comprehensively verify the convergence accuracy and stability of the algorithm while minimizing randomness, we conducted 50 independent runs of the GA, PSO, DBO, and TLK-DBO algorithms based on the studies of Ye et al. [28] and Yang et al. [29]. Taking into careful consideration their analysis of the population size and number of iterations, we set the population size to 50 and the number of iterations to 200 for each run. The average adaptation degree of 50 cycles and the change curve are used to react to the running accuracy and convergence speed of the algorithm. The running results and data comparison are shown in Table 7 and Fig 10.

After calculating through 50 cycles, $f_1 - f_8$ as shown in Table 7, the mean and standard deviation of the optimal fitness of TLK-DBO are greatly improved relative to GA, PSO, and DBO. As shown in Fig 10, TLK-DBO has higher solving accuracy and faster convergence speed relative to GA, PSO, and DBO, achieving optimal solutions in shorter iteration times. These results demonstrate that the TLK-DBO algorithm outperforms GA, PSO, and DBO in terms of faster convergence speed, higher convergence accuracy, and stronger stability.

## 4.2 Design of fuzzy controller for charging of traveling vehicles

The input signal needs to be fuzzified before the fuzzy controller is built, and then the fuzzy output is derived from the established fuzzy rule base. First of all, a reasonable scale transformation is carried out so that it can ensure that the driving demand of the car is satisfied, and then ensure that the driving charging mode can be carried out properly, and it is required that the thesis domain is in the appropriate range, $Q$ as shown in Eq (15). In the traveling charging mode, the demand torque is less than the economic torque $T_{e\_Opt}$, and its effective thesis domain is [0 1], the thesis domain of the battery $SOC$ is [0 0.5], while the range of the output is [0.3 1]. For $Q$ and $M$, we divide it into five fuzzy subsets {SL (minimum), L (small), Z (moderate), H (high), SH (high)}, while the battery $SOC$ is defined as {L (small), Z (moderate), H

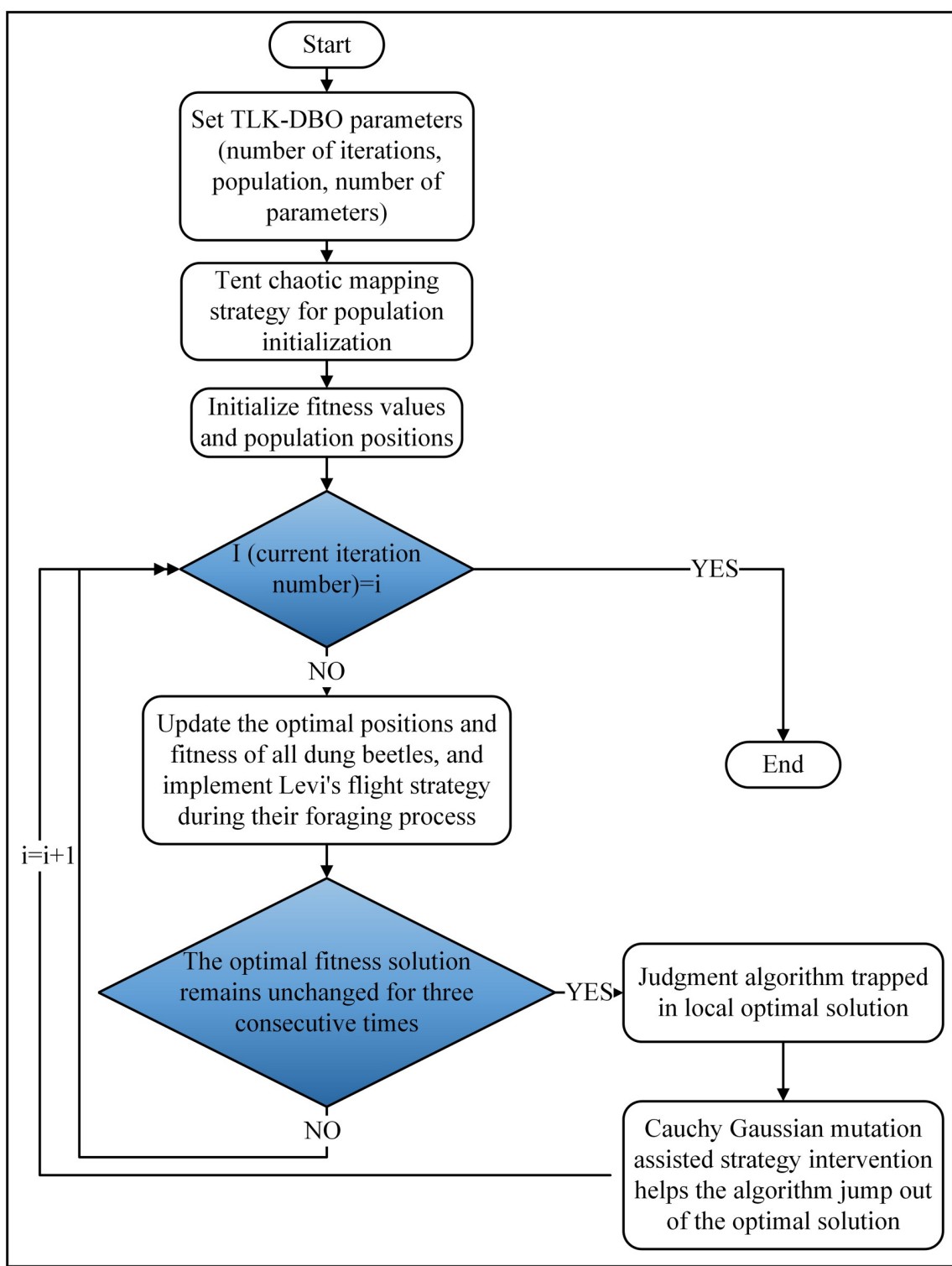

**Fig 9. Flowchart of the operation of TLK-DBO algorithm.**

**Table 6. Test function description table.**

| Function | Dim | Range | $f_{min}$ |
|---|---|---|---|
| $f_1(x) = 1 \sum_{i=1}^{n} x_i^2$ | 30 | [-100,100] | 0 |
| $f_2(x) = \sum_{i=1}^{n} |x_i| + \prod_{i=1}^{n} |x_i|$ | 30 | [-10,10] | 0 |
| $f_3(x) = \sum_{i=1}^{n} \left( \sum_{j=1}^{i} x_i \right)^2$ | 30 | [-100,100] | 0 |
| $f_4(x) = \sum_{i=1}^{n} (ix_i^4) + random[0,1]$ | 30 | [-1.28,1.28] | 0 |
| $f_5(x) = \frac{1}{4000} \sum_{i=1}^{n} x_i^2 - \prod_{i=1}^{n} cos\left(\frac{x_i}{\sqrt{i}}\right) + 1$ | 30 | [-600,600] | 0 |
| $f_6(x) = \left( \frac{1}{500} + \sum_{j=1}^{25} \frac{1}{j + \sum_{i=1}^{2} (x_i - a_{ij})^6} \right)^{-1}$ | 2 | [-65.536, -65.536] | 0.998003837 |
| $f_7(x) = \sum_{i=1}^{11} \left[ a_i - \frac{x_i(b_i^2 + b_i x_2)}{b_i^2 + b_i x_3 + x_4} \right]^2$ | 4 | [-5, -5] | 0.0003075 |
| $f_8(x) = - \sum_{i=1}^{10} \left[ (x - a_i)(x - a_i)^T + c_i \right]^{-1}$ | 4 | [0,10] | -10.5364 |

(high)}, whose fuzzy rule table is shown in Table 8. To carry out the design of the affiliation function, as shown in Fig 11.

$$Q = \frac{T_{req}}{T_{e_{Opt}}} \tag{15}$$

### 4.3 Hybrid drive fuzzy controller design

The hybrid driving mode defines the first input as $L$. $L$ is shown in Eq (16). The hybrid driving mode, where the demand torque $T_{req}$ is greater than the economic torque $T_{e\_Opt}$, has an effective range of [1.0 1.8], the battery $SOC$ has a range of [0.3 1], and the output has a range of [0.5 1]. The inputs and outputs are in the same form as the traveling charging mode. Its fuzzy rule table is shown in Table 9. The design of the affiliation function is carried out as shown in Fig 12.

$$L = \frac{T_{req}}{T_{e_{Opt}}} \tag{16}$$

Fuzzy control mostly relies on engineering experience, which cannot guarantee the control accuracy and global optimization results. In this paper, an optimization algorithm will be used

**Table 7. Comparison of experimental results.**

| Function | GA | | PSO | | DBO | | TLK-DBO | |
|---|---|---|---|---|---|---|---|---|
| | AVG | STD | AVG | STD | AVG | STD | AVG | STD |
| $f_1$ | 29734.35 | 6896.05 | 21.27 | 2.65 | 2.71E-34 | 1.91E-33 | 2.09E-42 | 1.47E-41 |
| $f_2$ | 71.34 | 9.46 | 19.69 | 1.98 | 4.09E-21 | 2.47E-20 | 4.92E-24 | 3.12E-23 |
| $f_3$ | 60924.76 | 22739.79 | 594.47 | 237.39 | 5.39E-12 | 3.81E-11 | 1.69E-25 | 8.06E-25 |
| $f_4$ | 25.93 | 14.18 | 101.43 | 20.25 | 3.06E-03 | 3.13E-03 | 2.73E-03 | 2.57E-03 |
| $f_5$ | 270.78 | 83.63 | 73.48 | 9.3 | 2.58E-04 | 1.82E-03 | 0 | 0 |
| $f_6$ | 1.62 | 1.34 | 2.04 | 1.44 | 1.59E+00 | 1.57E+00 | 1.04E+00 | 2.81E-01 |
| $f_7$ | 0.019395 | 0.021474 | 0.003625 | 0.0062566 | 8.76E-04 | 4.45E-04 | 3.79E-04 | 1.81E-04 |
| $f_8$ | -1.0587 | 0.37166 | -7.1773 | 1.9183 | -7.59E+00 | 3.34E+00 | -1.04E+01 | 7.65E-01 |

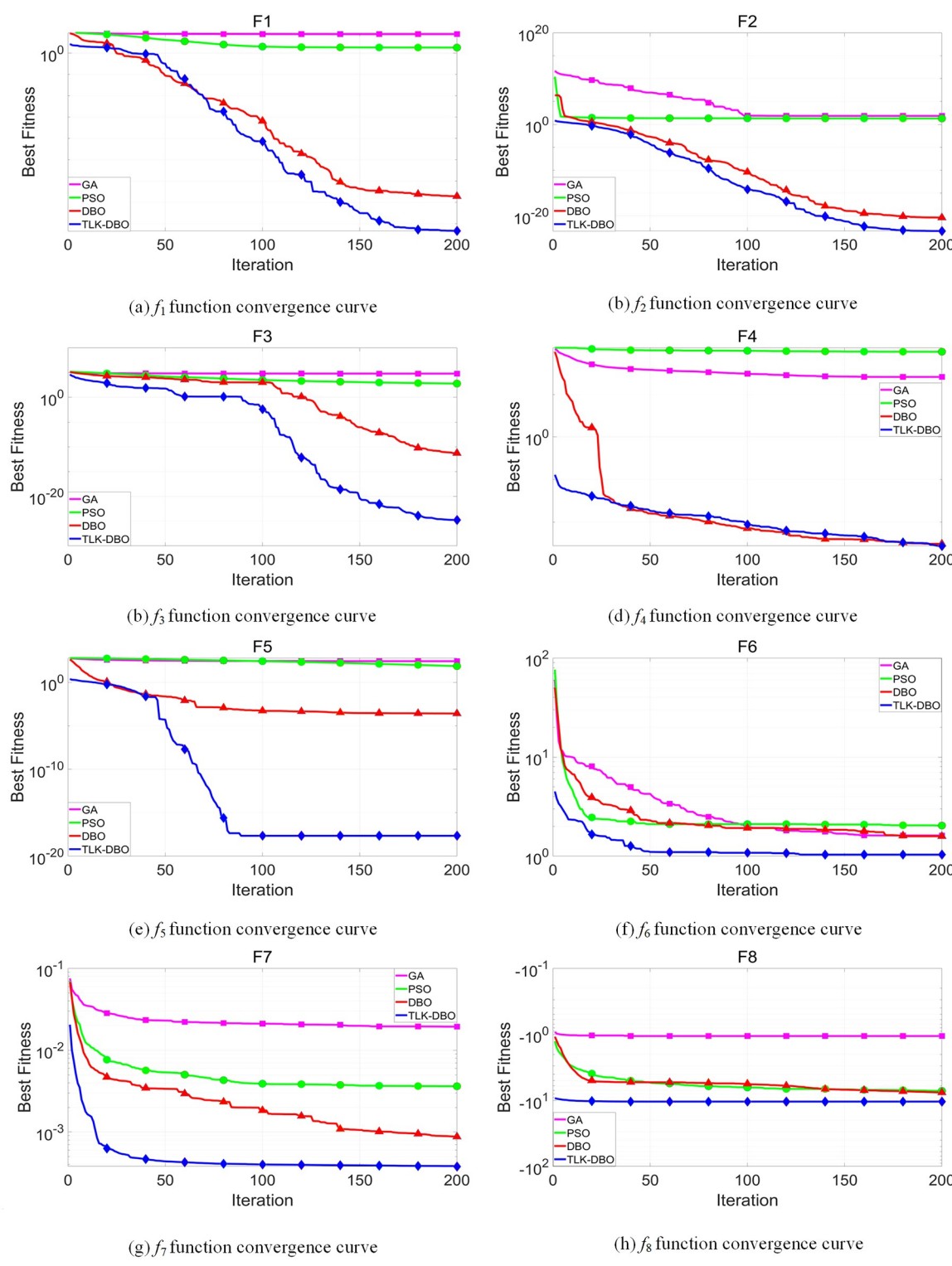

(a) $f_1$ function convergence curve

(b) $f_2$ function convergence curve

(b) $f_3$ function convergence curve

(d) $f_4$ function convergence curve

(e) $f_5$ function convergence curve

(f) $f_6$ function convergence curve

(g) $f_7$ function convergence curve

(h) $f_8$ function convergence curve

**Fig 10. Graph of operation results.**

**Table 8. Fuzzy rules for driving charging mode.**

| SOC | Q | | | | |
|---|---|---|---|---|---|
| | **SL** | **L** | **Z** | **H** | **SH** |
| L | Z | Z | H | SH | SH |
| Z | L | Z | Z | H | SH |
| H | SL | L | Z | H | H |

to find the optimal affiliation function in the set fuzzy controller, in order to find the optimal control effect. As an example, the inputs Q in the line charging mode are shown in Fig 13, $x_1$ ~ $x_{11}$ which are all the affiliation parameters that need to be optimized, and with the change of these parameters, the coordinates of the triangle and trapezoid are determined, which affect the output of the engine torque. The total number of parameters to be optimized for the affiliation function of the above two fuzzy controllers is 59.

On this basis, the fuzzy controller is optimized by the TLK-DBO optimization algorithm to reduce the methanol consumption and the variation of the battery *SOC* value as much as possible to obtain the maximum economic benefits and extend the life of the power battery. Therefore, the objective function expression is established as:

$$F(x) = \frac{\omega_1}{L_{Fuel}} \int Fuel(t)dt + \frac{\omega_2}{L_{SOC}} \int SOC(t)dt \tag{17}$$

Where, $\omega_1$, $\omega_2$ are the weight factors of the optimization objectives of engine methanol consumption and battery power, which take the values of 0.7,0.3, respectively;, $L_{Fuel}$, $L_{SOC}$ are the Methanol consumption and battery *SOC* before optimization, respectively. Setting the number of algorithmic iterations to 50 and the number of populations to 30, the affiliation function in the fuzzy controller is optimized by GA, PSO, DBO, TLK-DBO algorithm, respectively, and optimization process of the TLK-DBO algorithm is shown in Fig 14, with the same process applied to the other algorithms.

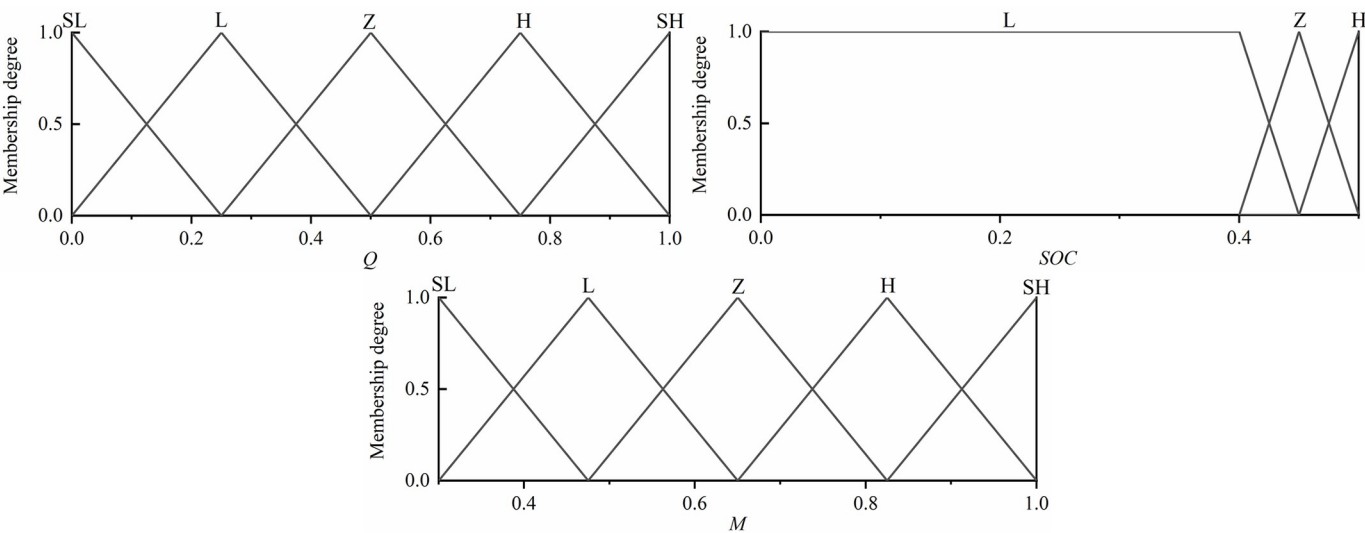

**Fig 11. Fuzzy membership degree of driving charging mode.**

**Table 9. Fuzzy rules for hybrid driving mode.**

| SOC | L | | | | |
|---|---|---|---|---|---|
| | **SL** | **L** | **Z** | **H** | **SH** |
| L | L | Z | Z | H | SH |
| Z | SL | L | Z | H | SH |
| H | SL | L | L | Z | H |

## 5 Comparative analysis of optimization results

In the actual driving process, when the battery *SOC* is lower than the target value, it is necessary to target the whole vehicle economy with small changes in battery *SOC* fluctuations for charging the vehicle to ensure the optimal economy of the whole vehicle. The changes of battery *SOC* values of the six strategies under the China Semi-trailer Tractor Cycle (CHTC-TT) conditions are shown by Fig 15. The initial values are all 40%, and the battery *SOC* at the end of Rule-EMS is 45.11%; the battery *SOC* at the end of Fuzzy-EMS, GA-Fuzzy-EMS, PSO-- Fuzzy-EMS, DBO-Fuzzy-EMS, TLK-DBO-Fuzzy-EMS are 45.21%, 45.15%, 44.76%, 44.22%, and 43.74%, respectively. The variations were 12.78%, 13.02%, 12.88%, 11.90%, 10.55%, and 9.35% for the six strategies, respectively.

Table 10 shows the methanol consumption under six different strategies. The methanol consumption in Rule-EMS is 121.75L/100km, Fuzzy-EMS, GA-Fuzzy-EMS, PSO-Fuzzy-EMS, DBO-Fuzzy-EMS, and TLK-DBO-Fuzzy-EMS reduce the methanol consumption by 3.89%, 4.45%,6.28%,7.73%, and 9.07%, respectively, under the same conditions It can be seen that the TLK-DBO-Fuzzy-EMS is able to find a better economy compared to the other controller.

Fig 16 shows the cumulative curve of methanol consumption for the engine under CHTC-TT conditions for the six strategies. Comparative analysis indicates that the TLK-DBO-Fuzzy-EMS strategy maximizes the overall vehicle economy relative to the other five strategies. Based on the methanol consumption and battery *SOC* changes, it can be inferred that, while ensuring optimal economy, reducing *SOC* fluctuations throughout the entire driving cycle helps minimize instability in internal chemical reactions and occurrences of electrochemical corrosion. This contributes to extending the battery's lifespan.

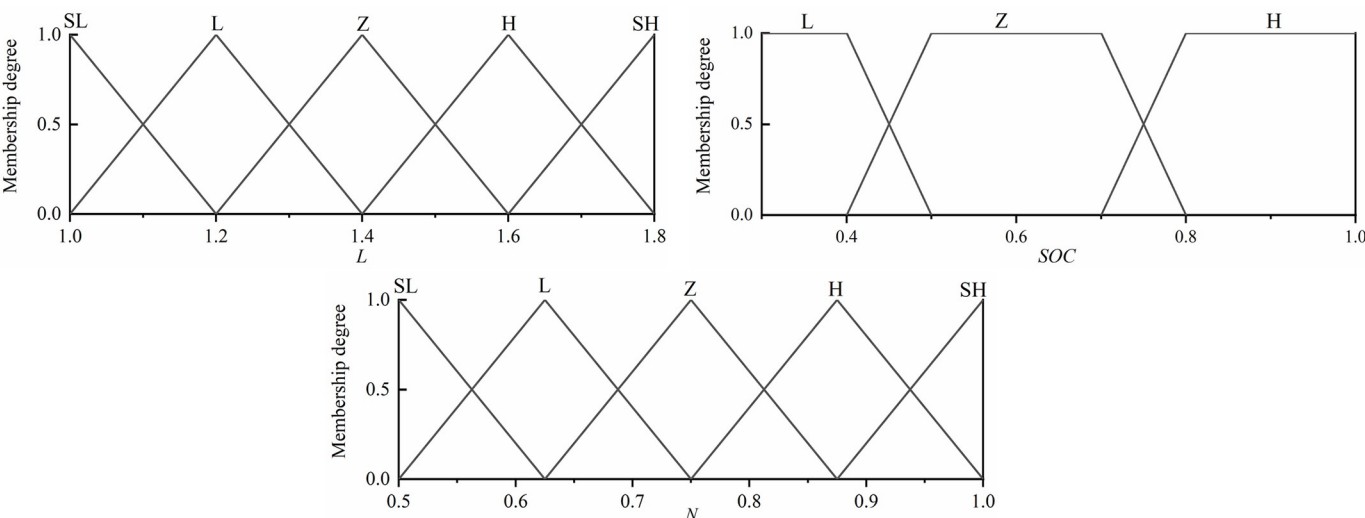

**Fig 12. Fuzzy membership degree of hybrid driving mode.**

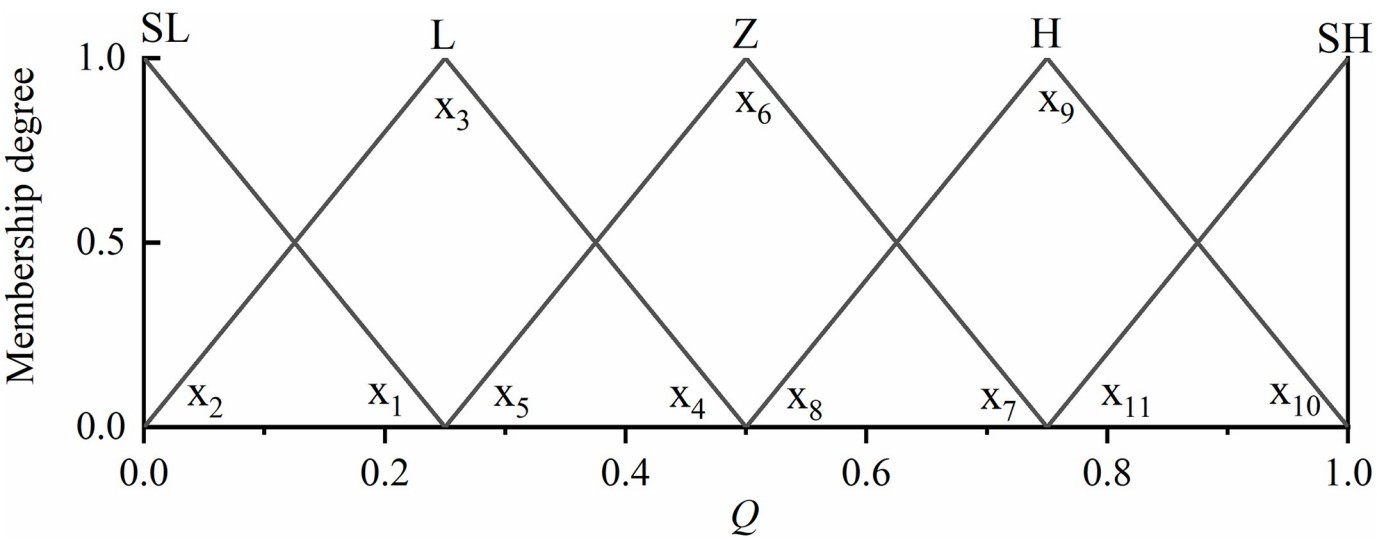

**Fig 13. Optimization parameters for input affiliation function.**

Fig 17 shows a comparison of the engine operating points under six strategies. As shown in the figure, the engine can operate within the delineated range. In Rule-EMS (Fig 17(A)), the engine operating point is relatively fixed. In Fuzzy-EMS (Fig 17(B)), it is evident that after fuzzy control adjustment, the engine operating point is closer to the high-efficiency zone. In

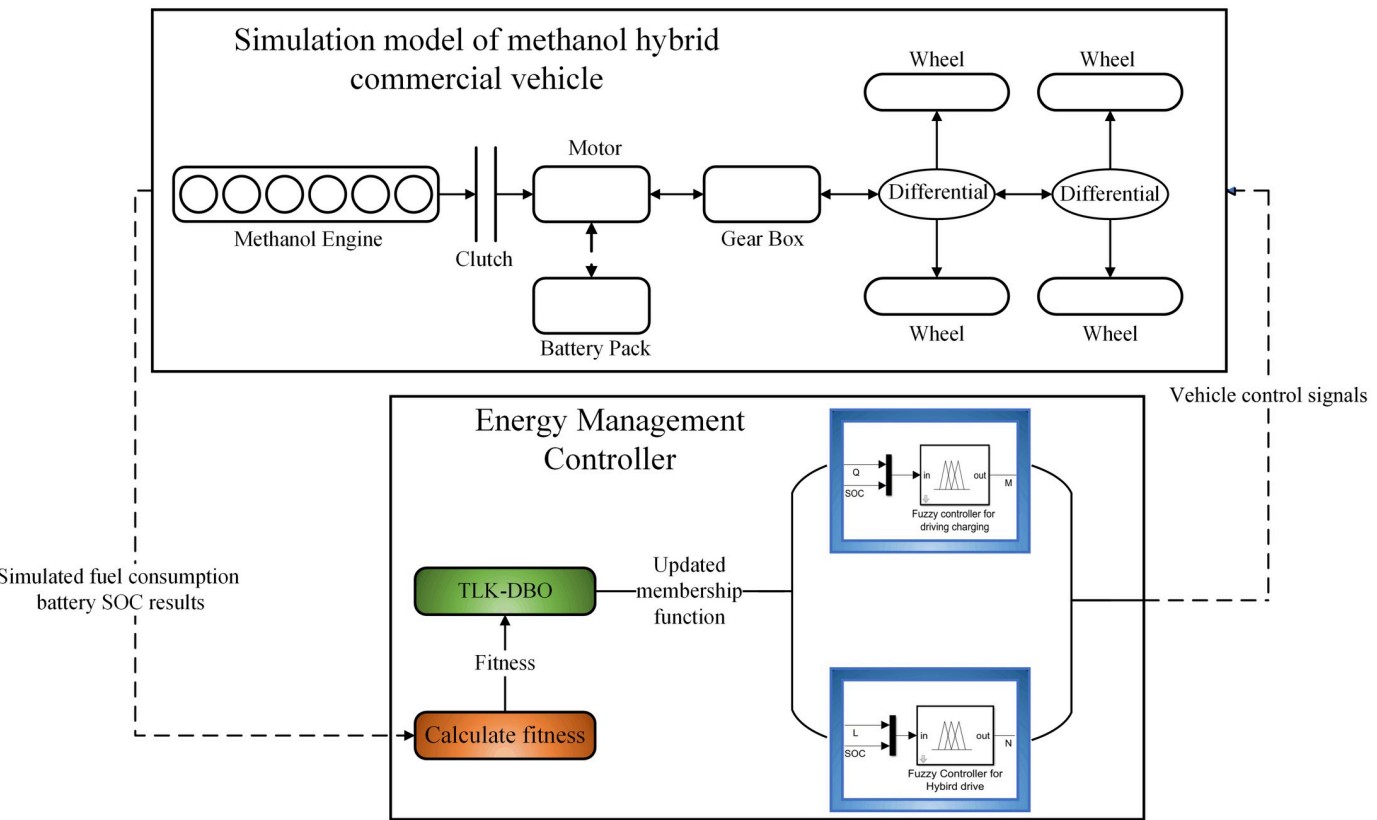

**Fig 14. Algorithm optimization process.**

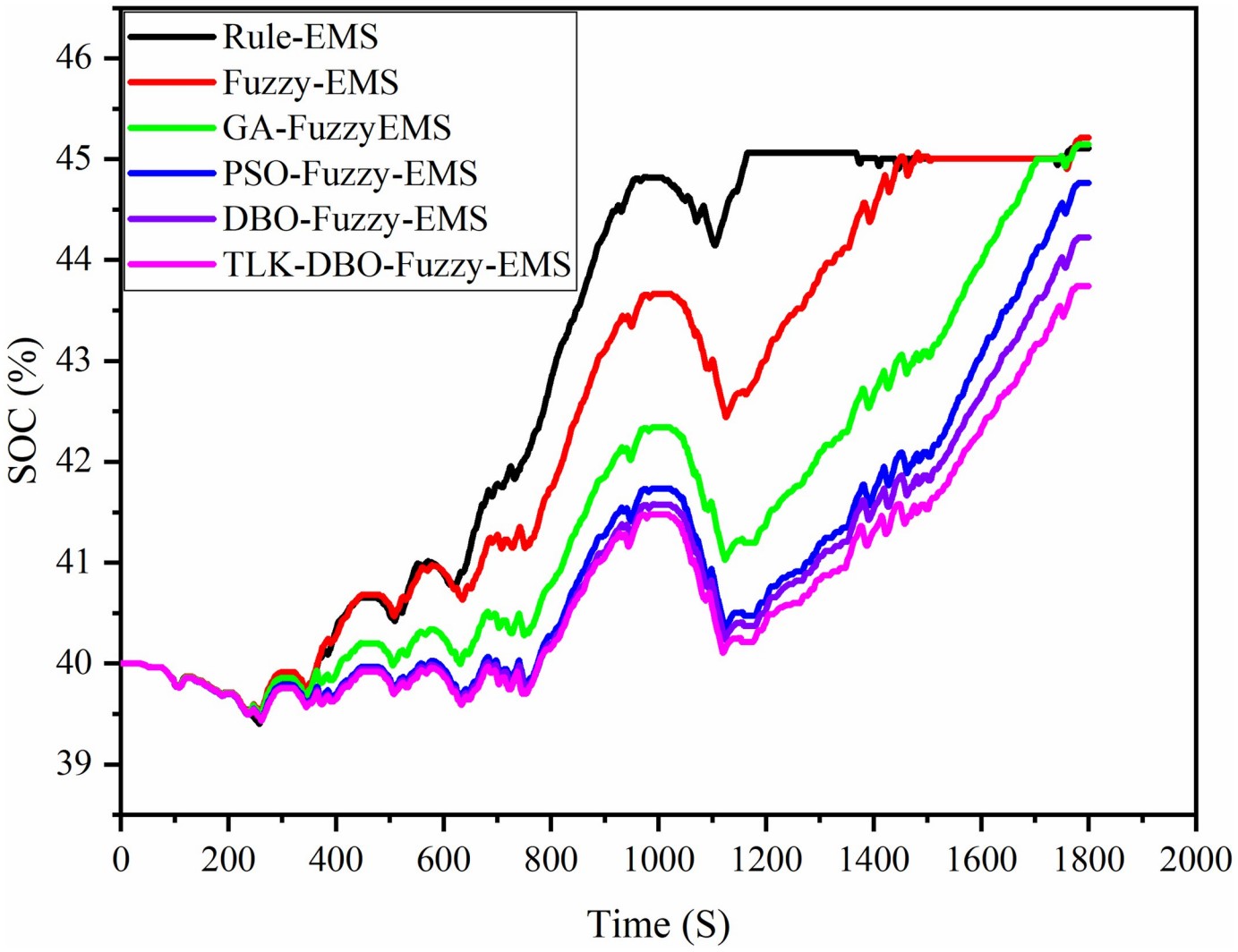

**Fig 15. Variation of battery *SOC* with six strategies.**

GA-Fuzzy-EMS (Fig 17(C)), PSO-Fuzzy-EMS (Fig 17(D)), DBO-Fuzzy-EMS (Fig 17(E)), and TLK-DBO-Fuzzy-EMS (Fig 17(F)), the optimized energy management strategies result in more engine operating points clustered in the high-efficiency zone. With the minimum torque set at $800Nm$, the engine operating points in the inefficient zone for the six strategies are

**Table 10. Simulation results of Methanol consumption of each energy management control strategy.**

| Driving cycle | EMS | Oil consumption |
|---|---|---|
| CHTC-TT | Rule | 121.75L/100km |
| | Fuzzy | 117.01L/100km |
| | GA-Fuzzy | 116.32L/100km |
| | PSO-Fuzzy | 114.10L/100km |
| | DBO-Fuzzy | 112.34L/100km |
| | TLK-DBO-Fuzzy | 110.71L/100km |

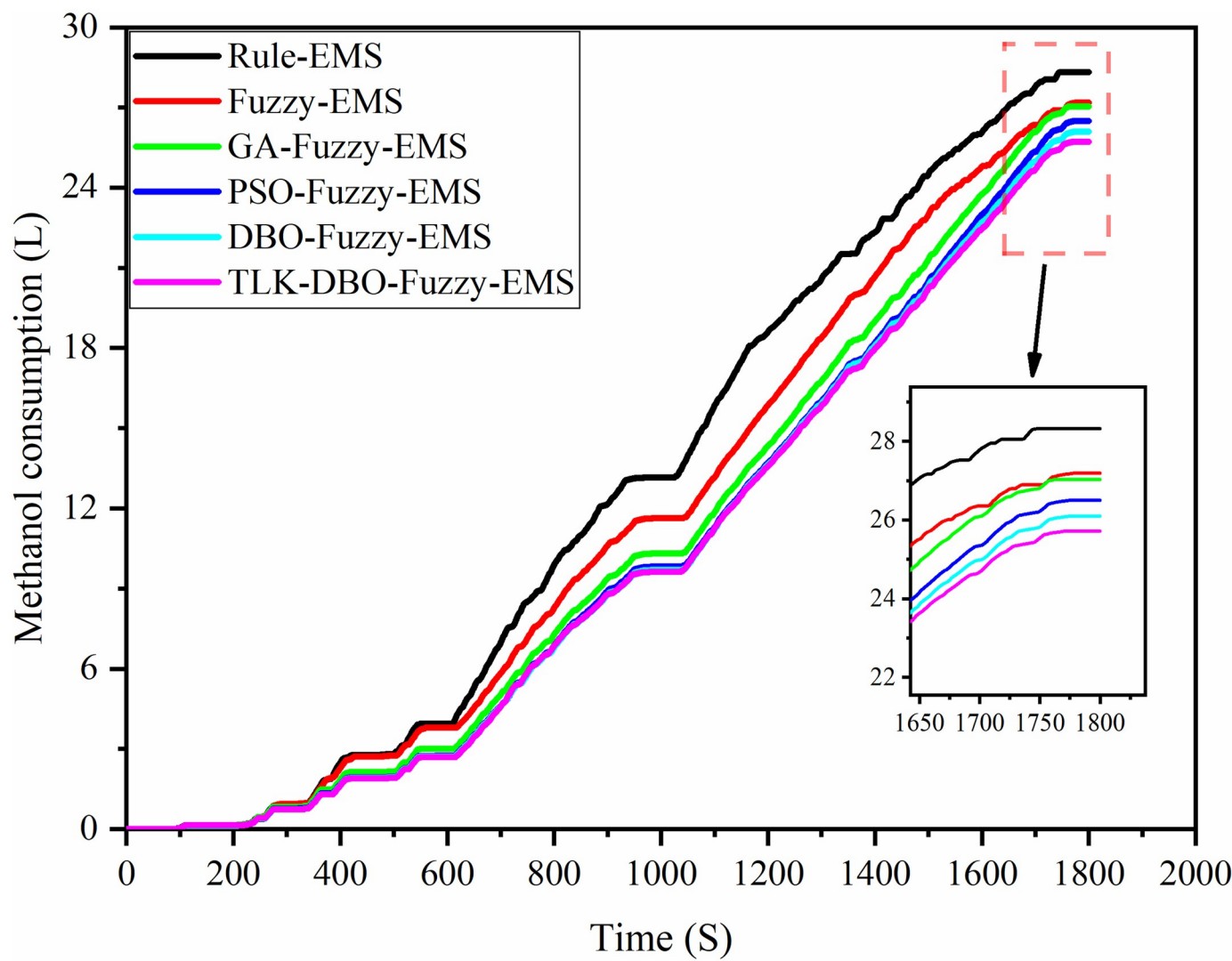

**Fig 16. Methanol consumption for six energy management strategies.**

34.42%, 25.10%, 16.46%, 17.27%, 19.55%, and 16.52%, respectively, as shown in the histogram in Fig 18. Although the engine operating points in the high-efficiency zone for TLK-DBO-Fuzzy-EMS and GA-Fuzzy-EMS are not significantly different, TLK-DBO-Fuzzy-EMS achieves the best methanol economy and reduces battery *SOC* fluctuations under the same conditions. It can be seen that TLK-DBO-Fuzzy-EMS can adjust the engine's output curve based on the real-time state of the vehicle during driving, showing greater robustness and adaptability compared to Rule-EMS. Compared to Fuzzy-EMS, it finds better fuzzy control strategies to further improve vehicle economy and reduce battery *SOC* fluctuations.

To further validate the universality and effectiveness of the proposed control strategy, we conducted tests using the China Heavy Truck Cycle (CHTC-HT) under various algorithmically optimized control strategies, as shown in Fig 19(A). The methanol consumption rates for the corresponding strategies were 144.68 L/km, 124.57 L/km, 118.98 L/km, 116.40 L/km, 115.76 L/km, and 115.07 L/km. The methanol consumption comparison results are depicted

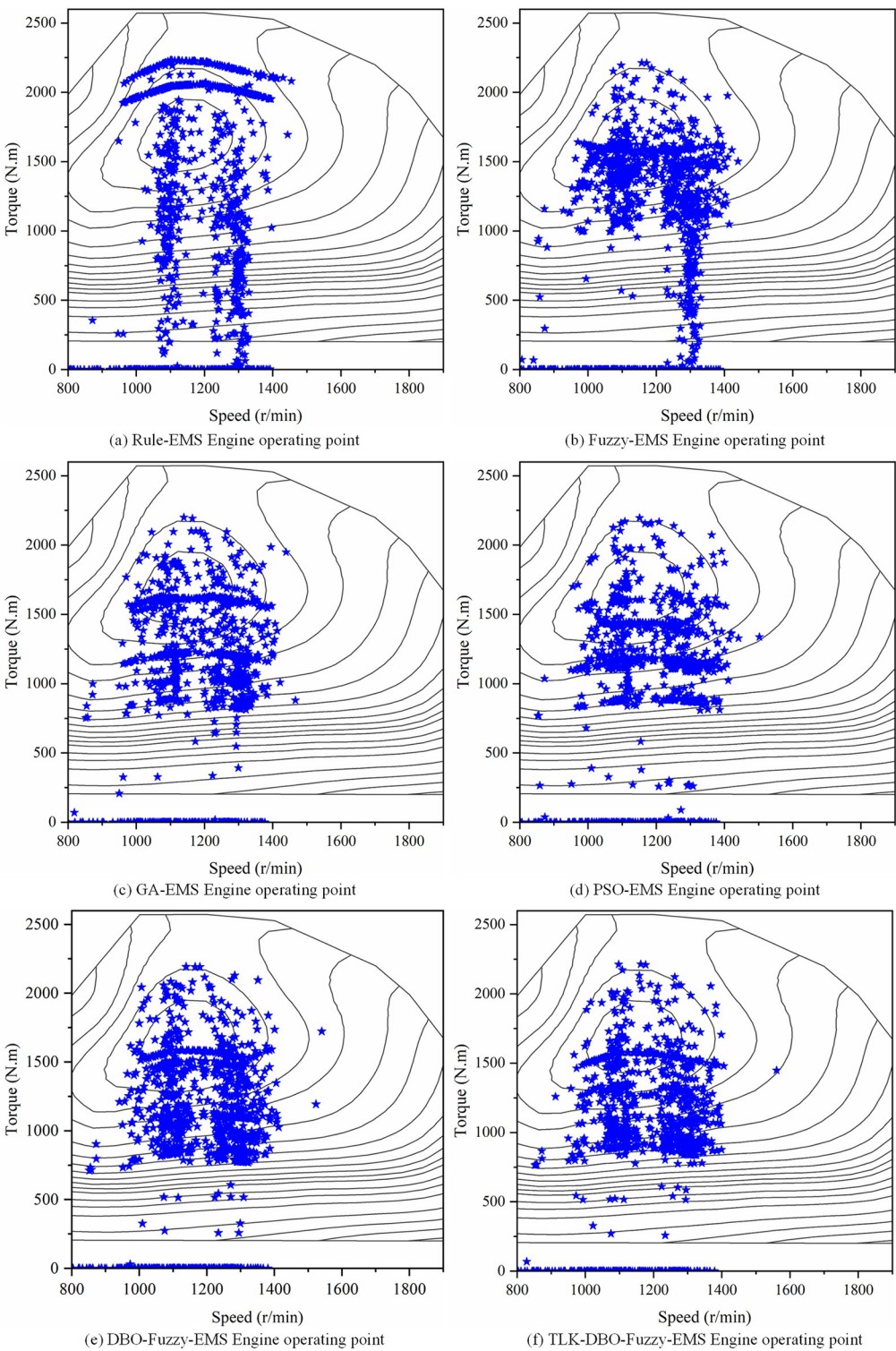

**Fig 17. Engine operating point for the six strategies.**

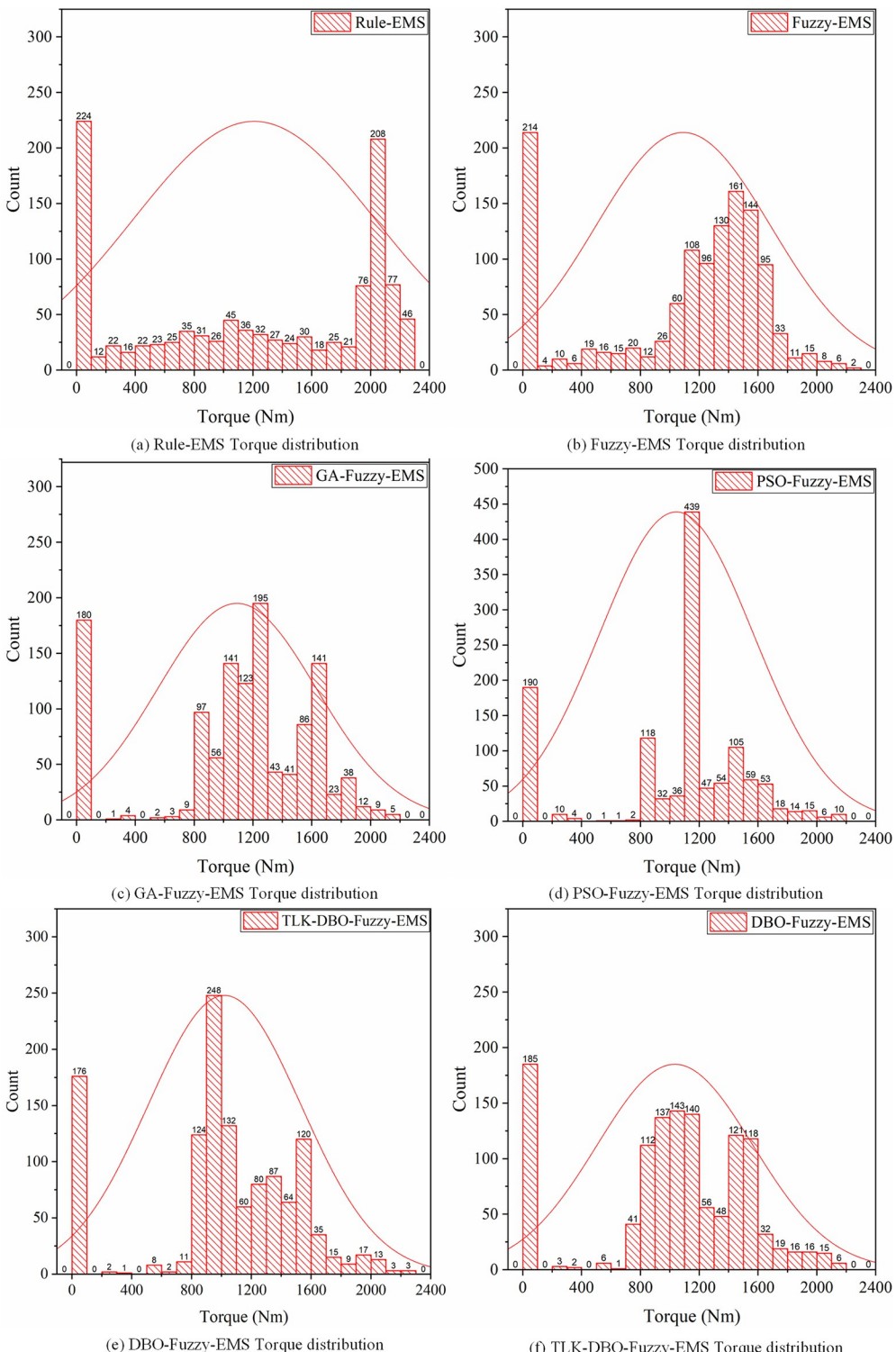

**Fig 18. Distribution of torque drop points under six strategies.**

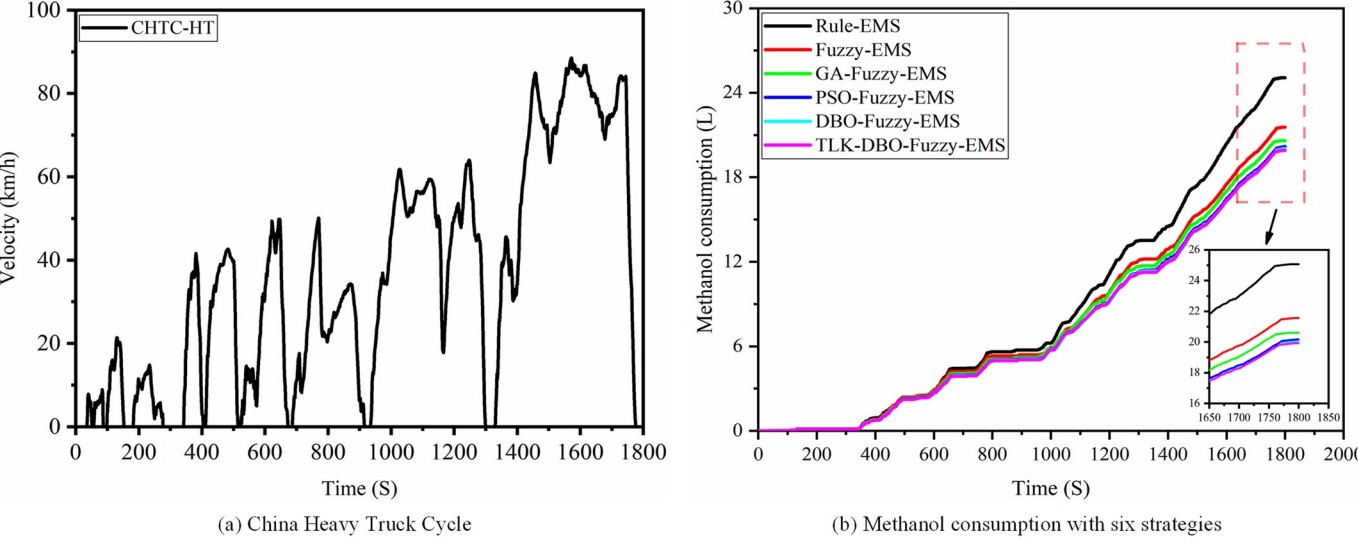

(a) China Heavy Truck Cycle

(b) Methanol consumption with six strategies

**Fig 19. China heavy truck cycle and methanol consumption.**

in Fig 19(B). The results verify that the improved dung beetle algorithm EMS has better economy and adaptability.

## 6 Concluding remarks

In this paper, in order to improve the adaptability and poor robustness of the rule-based EMS, a multi-fuzzy control EMS with improved optimization of DBO is proposed.

In order to improve the optimization effect of fuzzy control, this paper incorporates the population initialization strategy of Tent chaotic mapping fused with sinusoidal cosinusoidal random assignment, the dung beetle forager fused with Lévy flight strategy, and the improvement strategy of Cauchy's Gaussian variation on the basis of traditional DBO, which is shown to be able to greatly improve the global optimality searching ability, convergence speed, and optimality searching accuracy of the traditional DBO demonstrated through simulations on eight test functions. Unlike most scholars who only add a single fuzzy controller in drive mode, this paper incorporates fuzzy controllers in both linear charging and hybrid drive modes, allowing for a better division of the battery *SOC* working area in different modes and effectively addressing the issue of excess engine output power. Compared to the currently most widely used rule-based EMS, the improved DBO-optimized multi-fuzzy control in this paper reduces the overall methanol consumption of the vehicle by 9.07% and the fluctuation of the battery *SOC* by 3.43%, effectively enhancing the vehicle's economy, decreasing the fluctuation of the battery *SOC*, and greatly improving the adaptability and robustness issues of the rule-based EMS. Comparison of the optimization results proves that TLK-DBO has better optimization effect than the traditional DBO, and the engine working point is more in the delineated high-efficiency working area, which effectively improves the economy of the whole vehicle and reduces the fluctuation of the battery *SOC*. Meanwhile, this paper further validates the feasibility and effectiveness of the optimization method through different operating conditions.

## Author Contributions

**Funding acquisition:** Ping Xiao, Jiabao Pan.

**Resources:** Ping Xiao.

**Software:** Zhihao Li, Wenjun Pei, Aoning Lv.

**Supervision:** Ping Xiao, Jiabao Pan.

**Writing – original draft:** Zhihao Li, Wenjun Pei.

**Writing – review & editing:** Ping Xiao, Jiabao Pan.

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
