## [Decision Letter · Decision Letter 0]

4 Jun 2024

PONE-D-24-20196Energy Management Strategy for Methanol Hybrid Commercial Vehicles Based on Improved Dung Beetle Algorithm OptimizationPLOS ONE

Dear Dr. Xiao,

Thank you for submitting your manuscript to PLOS ONE. After careful consideration, we feel that it has merit but does not fully meet PLOS ONE’s publication criteria as it currently stands. Therefore, we invite you to submit a revised version of the manuscript that addresses the points raised during the review process.

We look forward to receiving your revised manuscript.

Kind regards,

Mazyar Ghadiri Nejad, Ph.D.

Academic Editor

PLOS ONE

Journal Requirements:

2. Please note that PLOS ONE has specific guidelines on code sharing for submissions in which author-generated code underpins the findings in the manuscript. In these cases, all author-generated code must be made available without restrictions upon publication of the work. 

Please review our guidelines at https://journals.plos.org/plosone/s/materials-and-software-sharing#loc-sharing-code and ensure that your code is shared in a way that follows best practice and facilitates reproducibility and reuse.

"The research is supported by Key Research and Development Projects in Anhui Province (2022a05020007) and the National Natural Science Foundation of China (52375227)."

Reviewers' comments:

Reviewer's Responses to Questions

**Comments to the Author**

1. Is the manuscript technically sound, and do the data support the conclusions?

Reviewer #1: Yes

Reviewer #2: Yes

2. Has the statistical analysis been performed appropriately and rigorously? 

Reviewer #1: Yes

Reviewer #2: Yes

3. Have the authors made all data underlying the findings in their manuscript fully available?

Reviewer #1: Yes

Reviewer #2: Yes

4. Is the manuscript presented in an intelligible fashion and written in standard English?

Reviewer #1: Yes

Reviewer #2: Yes

5. Review Comments to the Author

Reviewer #1: The manuscript proposes "Energy Management Strategy for Methanol Hybrid Commercial Vehicles Based on Improved Dung Beetle Algorithm Optimization".

The reviewer's concerns are as follows:

1- The novelty of this study is not clear. A flowchart summarizing the article should be added to the introduction section.

2- Authors have not provided sufficient comparison between their proposed methodology and the previous ones in the introduction. According to the current introduction, there is no contribution in the current study.

7- The reference numbers should be further increased. It is recommended to add the following reference.

a)Adak, S., Cangi, H., Kaya, R., Yılmaz, A. S. (2022). Effects of Electric Vehicles and Charging Stations on Microgrid Power Quality. Gazi University Journal of Science Part A: Engineering and Innovation, 9(3), 276-286. https://doi.org/10.54287/gujsa.1153313

Reviewer #2: The report presents the outcome of a study on the energy management strategy for methanol hybrid commercial vehicles. There is a need for the author(s) to improve the manuscript and the quality of the presentation using the comments and queries below:

1. It is pertinent to ask the author to update the manuscript with the answer to an important question like

1a) "How does the performance of the improved dung beetle algorithm compare to other optimization algorithms (such as genetic algorithms or particle swarm optimization) in terms of fuel efficiency and emissions in hybrid commercial vehicles?"

1b. How does the improved energy management strategy perform in real-world driving conditions, including various traffic scenarios and weather conditions, compared to controlled testing environments?

1c. What is the sensitivity of the improved dung beetle algorithm to different parameters such as the size of the population, number of iterations, and initial conditions?

1d. How does the improved energy management strategy affect the long-term durability and performance of the methanol engine and power battery? Are there any observed trade-offs between immediate fuel economy and long-term vehicle health?

2. Can you provide more detailed explanations and justifications for the choice of Tent chaotic mapping and the specific strategies (cosine, Lévy flight, and Cauchy Gaussian mutation) used to improve the dung beetle algorithm?

3. Does the report include a comprehensive comparison between the proposed energy management strategy and existing strategies in terms of efficiency, robustness, and adaptability? If not, can this comparison be added to highlight the improvements made?

4. Because of processing during production, could you please address all the in-text citation errors in the report? For instance

4a) Page 11 of 40: ... specific operating conditions. Literature [13-14] selects control parameters through dynamic

4b) Page 11 of 40: ... improve the overall vehicle economy. Literature [15] utilized dynamic programming in

4c) Page 11 of 40: ... strategies. Literature [17] used a dynamic programming algorithm to obtain the global optimum

Comment: It is important to note that it is absurd and improper to cite like [1],[2],[3],[4] or [1,2,3,4]. One way to reduce erroneous in-text citations, avoid mismatching in-text citations, and properly connect them to References is to insert the author's name (i.e., Aaagah [1] or Aaagah et al. [1]). Update each reference number with the author's name following the 6th Edition of APA style.

5. The manuscript, first paragraph of the introduction lack a fact on how heat transfer affects energy shortage. Could your check a suggestion below?

According to xcxccxxcc et al. (2022), the efficiency of heat transfer processes significantly impacts energy shortages by determining the effectiveness of energy utilization and conservation within a system, thereby influencing overall energy demand and supply dynamics.

Source: xcxccxxcc, I. L., xxxxx, N. A., xxxxx, A., xxxxx, B., fffff, R., & Koriko, O. K. (2022). Ratio of Momentum Diffusivity to Thermal Diffusivity: Introduction, Meta-analysis, and Scrutinization. Chapman and Hall/CRC. New York. ISBN-13: 978-1032108520, ISBN-10: 1032108525, ISBN9781003217374.

6. PLOS authors have the option to publish the peer review history of their article (what does this mean?). If published, this will include your full peer review and any attached files.

Reviewer #1: No

Reviewer #2: No

---

## [Author Response · Author response to Decision Letter 0]

9 Jul 2024

Journal Requirements

1. Please ensure that your manuscript meets PLOS ONE's style requirements, including those for file naming. The PLOS ONE style templates can be found at two PDF.

Response: Thank you for your feedback. We have carefully reviewed both PDF files and made the necessary revisions throughout the manuscript as requested. For images that were modified, we indicated the changes in red font. Additionally, all images have been uploaded in TIFF format. (Use blue markings for formatting changes.)

2. Please note that PLOS ONE has specific guidelines on code sharing for submissions in which author-generated code underpins the findings in the manuscript. In these cases, all author-generated code must be made available without restrictions upon publication of the work.

Response: Due to proprietary improvements in the algorithm, we regret that we cannot publicly share the code related to our research. However, we are committed to transparency and reproducibility in our research. For those who wish to obtain the code, please contact us via email at (2220110119@stu.ahpu.edu.cn). We will promptly provide the code and any necessary documentation to ensure a thorough evaluation of our manuscript.

Response: Thank you for pointing out our oversight and correcting our mistake. We will carefully review our funding information during the resubmission process.

4. Thank you for stating the following financial disclosure: "The research is supported by Key Research and Development Projects in Anhui Province (2022a05020007) and the National Natural Science Foundation of China (52375227)." Please state what role the funders took in the study. If the funders had no role, please state: "The funders had no role in study design, data collection and analysis, decision to publish, or preparation of the manuscript."

If this statement is not correct you must amend it as needed. Please include this amended Role of Funder statement in your cover letter; we will change the online submission form on your behalf.

Response: Thank you for your correction. We will re-upload the cover letter and clarify the contributions made by the funders in the manuscript with the red-marked sections. We appreciate your assistance in making these changes.

The specific content is “The research was supported by Key Research and Development Projects in Anhui Province (2022a05020007) and the National Natural Science Foundation of China (52375227). Professors Ping Xiao and Jiabao Pan made substantial contributions to the research direction design, data analysis, and manuscript revisions.” (In the re-uploaded cover letter, mark changes in blue font.)

Response: Thank you for your suggestion. We have thoroughly checked the cited references, corrected their formatting, and marked the added references in red font. We have also renumbered them accordingly in red font. If you have any further suggestions, we would greatly appreciate your feedback. (Please see the modifications in the "References" section.)

Reviewer: 1

1.The novelty of this study is not clear. A flowchart summarizing the article should be added to the introduction section.

Response: Thank you for your valuable suggestions! We take your constructive comments and make the novelty of the article clearer, the structure of the article is introduced in the last part of the introduction and a flowchart summarizing the article has been added. (see line 113-122 and Fig 1 on page 6)

2. Authors have not provided sufficient comparison between their proposed methodology and the previous ones in the introduction. According to the current introduction, there is no contribution in the current study.

Response: Thank you to the reviewers for pointing out the shortcomings in our manuscript. Based on your suggestions, we have added detailed descriptions in the introduction to highlight the differences between our proposed method and previous methods. As follows, we have listed the main innovations of our work:

(1) Compared to hybrid assemblies using conventional diesel engines, this paper uses a more environmentally friendly and economically efficient methanol engine. (see line 97-98 on page 5)

(2) In the traditional method, only a fuzzy control is added to regulate the charging mode and the hybrid drive mode, but the efficient operating range of the engine is not the same in these two modes, which may reduce the effect of the fuzzy control and fail to achieve the best economy. In this paper, multi-fuzzy control is added to these two modes, which can enable the engine to fully utilize the optimum economy of the vehicle in different modes of operation. We have followed reviewer’s comments and compared our work with previous approaches in more detail in the introduction. (see line 98-101 on page 5)

(3) To develop a more effective energy management strategy, we employ the Dung Beetle Optimization (DBO) algorithm to optimize the fuzzy controller. However, the standard DBO algorithm often falls into local optima and converges slowly. To address these issues, we propose an improved DBO algorithm that enhances solution speed and accuracy. This improved DBO algorithm is then used to derive a more efficient energy management strategy. (see line 101-108 on page 5 and line 109-112 on page 6)

3.The reference numbers should be further increased. It is recommended to add the following reference.

Response: We sincerely thank the reviewers for their valuable and helpful comments, and based on your suggestions, we have added a narrative on the impact of hybrid vehicles on microgrids in the second part with references to the literature recommended by you.

The specific content is “Hybrid electric vehicles can recharge the power battery when the battery SOC falls below a set value, maintaining the balance of the battery SOC. This can reduce the dependence of hybrid vehicles on charging stations during operation and minimize the negative impact on the microgrid.”

(see line 130-133 on page 7 and reference 22)

Reviewer: 2

1. It is pertinent to ask the author to update the manuscript with the answer to an important question like：

1a) "How does the performance of the improved dung beetle algorithm compare to other optimization algorithms (such as genetic algorithms or particle swarm optimization) in terms of fuel efficiency and emissions in hybrid commercial vehicles?"

Response: Thank you very much for your suggestions. Your suggestions are very constructive for this article. Based on your suggestions, we have made the following improvements.

(1) We have added a comparison of the optimization speed and accuracy between the Genetic Algorithm (GA), Particle Swarm Optimization (PSO) algorithm, Dung Beetle Optimization (DBO) algorithm, and the improved Dung Beetle Optimization (TLK-DBO) algorithm in the manuscript. (see line 310-311 on page 17, line 320-329 on page 18, line 330-331 on page 19 and Fig 10, Table 7).

(2) We have added the energy management strategies optimized by the GA and PSO algorithms, and included an analysis of the battery SOC value changes for these two algorithms compared with other energy management strategies. (see line 381-383 on page 21, line 384-387 and Fig 15)

(3) Subsequently, we included the analysis of methanol consumption for the energy management strategies optimized by the GA and PSO algorithms compared with other energy management strategies. (see line 388-396 on page 22 and Fig 16, Table 10)

(4) The final analysis compared the operating points of the engine under six strategies. (see line 403-420 on page 23, line 421-426 on page 24 and Fig 17, Fig 18)

From the comparison, it is evident that the improved Dung Beetle Algorithm achieves better global optimum economy and effectively reduces battery SOC fluctuations, thereby concentrating the engine operating points more closely within the high-efficiency zone.

1b) How does the improved energy management strategy perform in real-world driving conditions, including various traffic scenarios and weather conditions, compared to controlled testing environments?

Response: Thank you very much for your suggestion. We have included the typical operating conditions of trucks in China to further validate the effectiveness of our proposed energy management strategy. And we have added the working condition graph and the comparison graph of then consumption under this working condition.

The specific content is “To further validate the universality and effectiveness of the proposed control strategy, we conducted tests using the China Heavy Truck Cycle (CHTC-HT) under various algorithmically optimized control strategies, as shown in Fig 19(a). The fuel consumption rates for the corresponding strategies were 144.68 L/km, 124.57 L/km, 118.98 L/km, 116.40 L/km, 115.76 L/km, and 115.07 L/km. The fuel consumption comparison results are depicted in Fig 19(b). The results verify that the improved dung beetle algorithm energy management strategy has better economy and adaptability” (see line 427-435 on page 24 and Fig 19)

“Meanwhile, this paper further validates the feasibility and effectiveness of the optimization method through different operating conditions.”(see line 454-455 on page 25)

1c) What is the sensitivity of the improved dung beetle algorithm to different parameters such as the size of the population, number of iterations, and initial conditions?

Response: We sincerely thank the reviewers for their feedback. The choice of population size, number of iterations and initial conditions in this paper is based on Ye M. and Yang G.L. who analyzed the effects of population size and number of iterations when improving the algorithm. After careful consideration, we chose a population size of 50 and 200 iterations to successfully verify the superior performance of the improved Dung Beetle Optimization (DBO) algorithm compared to Genetic Algorithm (GA), Particle Swarm Optimization (PSO), and traditional DBO algorithms.

The specific content is “To comprehensively verify the convergence accuracy and stability of the algorithm while minimizing randomness, we conducted 50 independent runs of the GA, PSO, DBO, and TLK-DBO algorithms based on the studies of Ye M et al. [28] and Yang GL et al. [29]. Taking into careful consideration their analysis of the population size and number of iterations, we set the population size to 50 and the number of iterations to 200 for each run.”

We appreciate the valuable feedback provided by the reviewers and hope this explanation adequately addresses your concerns. (see line 311-315 on page 17 and reference 28,29)

1d) How does the improved energy management strategy affect the long-term durability and performance of the methanol engine and power battery? Are there any observed trade-offs between immediate fuel economy and long-term vehicle health?

Response: Thank you very much for your meaningful feedback. In optimizing the energy management strategy, our objective was to achieve the optimal strategy that minimizes overall vehicle fuel consumption and reduces battery State of Charge (SOC) fluctuations. Ultimately, the energy management strategy we obtained improves the vehicle's overall fuel efficiency while also decreasing battery SOC variation. Reducing fluctuations in battery SOC helps extend battery life, thereby enhancing the overall lifespan of the vehicle.

The specific content is “Comparative analysis indicates that the TLK-DBO-Fuzzy-EMS strategy maximizes the overall vehicle economy relative to the other five strategies. Based on the methanol consumption and battery SOC changes, it can be inferred that, while ensuring optimal economy, reducing SOC fluctuations throughout the entire driving cycle helps minimize instability in internal chemical reactions and occurrences of electrochemical corrosion. This contributes to extending the battery's lifespan.” (line 396-398 on page 22 and line 399-401 on page 23)

2. Can you provide more detailed explanations and justifications for the choice of Tent chaotic mapping and the specific strategies (cosine, Lévy flight, and Cauchy Gaussian mutation) used to improve the dung beetle algorithm?

Response: We thank the jury experts for their corrections and apologize for the lack of explanation of our strategy for improving the algorithm. We have explained our reasons for choosing the improved strategy by revising the manuscript to describe the advantages of the improved strategy in more detail.

(1) Tent chaotic mappings are characterized by extensive randomness, continuity and smoothing, as well as enhanced diversity. Together, these properties enable the initial population to cover the entire search space uniformly, avoiding premature convergence to local optimal solutions and ensuring that the search process proceeds smoothly, balancing the needs of exploration and exploitation, thus improving the global search capability and robustness of the algorithm.Additionally, this paper combines Tent chaotic mapping with sine and cosine random assignment strategies, leveraging the advantages of chaotic and trigonometric mappings to further optimize population initialization. The sine and cosine strategies contribute to the breadth and smoothness of the search process, ensuring not only diversity within the population but also uniform distribution across the search space. (see line 243-256 on page 14 and reference 25)

(2) Lévy flights utilize their long-tailed distribution to effectively prevent algorithms from falling into local optima, thereby enhancing their ability to discover global optimal solutions. They can generate large step sizes, accelerating the algorithm's exploration of the entire search space and improving the efficiency of finding optimal solutions to optimization problems. This strategy is suitable for various complex and diverse optimization scenarios, showing strong adaptability without being limited by specific problem types. Additionally, Lévy flights maintain stable search performance and excel in handling complex, high-dimensional, and nonlinear problems, thereby enhancing the algorithm's robustness and practicality. (see line 271-279 on page 15 and reference 26)

(3) The Cauchy Gaussian mutation strategy utilizes the randomness of the Cauchy distribution to effectively enhance the algorithm's global search capability and robustness. It allows the algorithm to jump to distant positions, avoiding the risk of premature convergence to local optimal solutions, thereby enabling greater exploration and variability. This strategy enhances the algorithm's exploratory and diverse capabilities in complex, high-dimensional, and nonlinear optimization problems, significantly increasing the probability of finding global optimal solutions. Moreover, it is versatile and practical, as it is not limited by specific problem types or parameter settings. (see line 285-296 on page 16 and reference 27)

3. Does the report include a comprehensive comparison between the proposed energy management strategy and existing strategies in terms of efficiency, robustness, and adaptability? If not, can this comparison be added to highlight the improvements made?

Response: Thank you very much for your valuable suggestions. We have taken your advice and added a further explanation of the superiority of the energy management strategy proposed in this paper compared to rule-based and fuzzy control-based energy management strategies. The energy management strategy proposed in this paper has better adap

---

## [Decision Letter · Decision Letter 1]

24 Sep 2024

PONE-D-24-20196R1Energy Management Strategy for Methanol Hybrid Commercial Vehicles Based on Improved Dung Beetle Algorithm OptimizationPLOS ONE

Dear Dr. Ping Xiao,

Thank you for submitting your manuscript to PLOS ONE. After careful consideration, we feel that it has merit but does not fully meet PLOS ONE’s publication criteria as it currently stands. Therefore, we invite you to submit a revised version of the manuscript that addresses the points raised during the review process.

As you may see, the first and the third reviewers agreed to publish your manuscrıpt, but still there are some comments from the second reviewer that must be met. Hence, I suggest you to revise your manuscript considering the comments.If you have received any comment related to discuss closely about related literature and cite additional sources from reviewers, such suggestions should be considered as optional.

We look forward to receiving your revised manuscript.

Kind regards,

Mazyar Ghadiri Nejad, Ph.D.

Academic Editor

PLOS ONE

Journal Requirements:

Reviewers' comments:

Reviewer's Responses to Questions

**Comments to the Author**

1. If the authors have adequately addressed your comments raised in a previous round of review and you feel that this manuscript is now acceptable for publication, you may indicate that here to bypass the “Comments to the Author” section, enter your conflict of interest statement in the “Confidential to Editor” section, and submit your "Accept" recommendation.

Reviewer #1: (No Response)

Reviewer #2: All comments have been addressed

Reviewer #3: (No Response)

2. Is the manuscript technically sound, and do the data support the conclusions?

Reviewer #1: Yes

Reviewer #2: Yes

Reviewer #3: Yes

3. Has the statistical analysis been performed appropriately and rigorously? 

Reviewer #1: Yes

Reviewer #2: Yes

Reviewer #3: Yes

4. Have the authors made all data underlying the findings in their manuscript fully available?

Reviewer #1: Yes

Reviewer #2: Yes

Reviewer #3: Yes

5. Is the manuscript presented in an intelligible fashion and written in standard English?

Reviewer #1: Yes

Reviewer #2: Yes

Reviewer #3: Yes

6. Review Comments to the Author

Reviewer #1: (No Response)

Reviewer #2: The report presents the outcome of a study on Energy Management Strategy for Methanol Hybrid Commercial Vehicles Based on Improved Dung Beetle Algorithm Optimization. It is a necessary and sufficient condition that the report must be known for something significant. For this report to be recommended for acceptance, there is a need to address the technical issues and the minor comments.

IMPORTANT INFORMATION: Delete the old version of the manuscript from the submission portal.

1. Start each paragraph with the definition of term or keyword it tends to announce.

2. Could you please revise the introduction so that each paragraph starts with the definition of the keyword it tends to introduce?

3. Revise the introduction such that each paragraph starts with the definition of the keyword that it tends to introduce.

Comment: There is a need to restructure the introduction section; concept definition, theoretical review, and empirical review could be easily fetched. Note that each paragraph should announce at least an important study concept. Recall that the major components of a standard paragraph under the introduction are (a) The definition of the term/concept/idea is needed to understand the exact contribution of the report to the body of knowledge. (b) The theoretical review tells us about the published aim and significant theory. (c) The empirical review tells us about published results within the scope of the subject matter.

4. The abstract lacks clear separation between methods and results. Could you please separate the description of the methodology from the results for clarity. This helps readers understand the sequence of steps taken in the research.

5. It was written::

First, the rule-based energy management strategy is established by dividing the efficient working areas of methanol engine and power battery. Secondly, this paper employs Tent chaotic mapping to integrate the strategies of cosine, Lévy flight, and Cauchy Gaussian mutation, thus improving the dung beetle algorithm. This helps compensate for the traditional dung beetle algorithm's tendency to fall into local optima and enhances the algorithm's global search capability. And the fuzzy controllers for driving charging mode and hybrid driving mode are designed under the rule-based energy management strategy.

Could you revise or consider replacement with::

First, the rule-based energy management strategy is established by dividing the efficient working areas of the methanol engine and power battery. The Tent chaotic mapping is then used to integrate strategies of cosine, Lévy flight, and Cauchy Gaussian mutation, improving the dung beetle algorithm. This integration compensates for the traditional dung beetle algorithm's tendency to fall into local optima and enhances its global search capability. Subsequently, fuzzy controllers for the driving charging mode and hybrid driving mode are designed under this rule-based energy management strategy.

6. Line 31 - 35, it was written:

Hybrid electric vehicles can optimize energy management and thermal management systems to enhance heat transfer efficiency within the vehicle's energy transmission system. According to Animasaun, I.L et al. [1] the efficiency of heat transfer processes significantly impacts energy shortages by determining the effectiveness of energy utilization and conservation within a system, thereby influencing overall energy demand and supply dynamics.

Comment: Could you please revise and make it more concise? You may consider the replacement below:

Hybrid electric vehicles (HEVs) can optimize energy management and thermal management systems to enhance heat transfer efficiency within the vehicle's energy transmission system. According to Animasaun et al. [1], the efficiency of heat transfer processes plays a crucial role in addressing energy shortages. Efficient heat transfer determines the effectiveness of energy utilization and conservation within a system, directly influencing overall energy demand and supply dynamics.

7. Line 52, it was written: Jeoung H et al. [8]obtains the optimal logic

Comment: Revise to: Jeoung et al. [8]obtains the optimal logic

8. Line 74, it was written: ... develop control rules. Anselma PG et al.[17] added the slope weighting method in dynamic...

Comment: Revise to:

... develop control rules. Anselma et al.[17] added the slope weighting method in dynamic...

9. Validation of the new results seems necessary.

Comment: The discussion of results must not stand in isolation. The authors need to compare the new results in the report with the existing related results in the literature. Consider how the results of the research article align with or differ from previous studies in the field. Discuss any similarities or contradictions and offer possible explanations for the differences. This is the best way to reduce the gap between the introduction and discussion of results.

Reviewer #3: (No Response)

7. PLOS authors have the option to publish the peer review history of their article (what does this mean?). If published, this will include your full peer review and any attached files.

Reviewer #1: No

Reviewer #2: No

Reviewer #3: No

---

## [Author Response · Author response to Decision Letter 1]

2 Oct 2024

Li, Zhihao

1.Anhui Province Key Laboratory of Intelligent Car Wire-Controlled Chassis System, Anhui Polytechnic University, Wuhu 241000, China

2.School of Mechanical and Automotive Engineering, Anhui Polytechnic University, Wuhu 241000, China

2220110119@stu.ahpu.edu.cn

Xiao, Ping (Corresponding Author)

1.Anhui Province Key Laboratory of Intelligent Car Wire-Controlled Chassis System, Anhui Polytechnic University, Wuhu 241000, China

2.School of Engineering, University of Brigeport, CT 06604, USA

tlxp95@ahpu.edu.cn

13866376423

Pan, Jiabao

1.Anhui Province Key Laboratory of Intelligent Car Wire-Controlled Chassis System, Anhui Polytechnic University, Wuhu 241000, China

2.School of Mechanical and Automotive Engineering, Anhui Polytechnic University, Wuhu 241000, China

panjiabao@ahpu.edu.cn

Pei, Wenjun

1.Anhui Province Key Laboratory of Intelligent Car Wire-Controlled Chassis System, Anhui Polytechnic University, Wuhu 241000, China

2.School of Mechanical and Automotive Engineering, Anhui Polytechnic University, Wuhu 241000, China

2220110151@stu.ahpu.edu.cn

Lv, Aoning

1.School of Mechanical and Automotive Engineering, Anhui Polytechnic University, Wuhu 241000, China

18110850667@163.com

Dear Editors,

Thank you very much for your supervision of the reviewing process of our manuscript with the reference number of PONE-D-24-20196. We also highly appreciate the reviewers’ and editors’ carefulness, conscientiousness, and the broad knowledge on the relevant research fields, since they have given us a number of beneficial suggestions. According to the reviewers’ and editors’ criticism and instruction, we have made the following revisions in terms of format and contents (which are marked in red or blue letters in manuscript):

Journal Requirements

Response: Thank you for your suggestion, we have thoroughly checked the references cited and made sure they are complete and correct. If you have any further suggestions, we would be grateful!

Reviewer: 1(No Response)

Reviewer: 2

1.Start each paragraph with the definition of term or keyword it tends to announce.

Response: Thank you for pointing out the shortcomings, we have revised the paragraphs starting with the missing keywords and explained them. Please contact us if you have any other suggestions.

The specific content is “Methanol, which has a higher content of oxygen atoms, is more easily and completely combusted, resulting in lower emissions [23].” (see line 173-174 on page 9)

The specific content is “The energy management strategy for hybrid vehicles aims to keep the engine operating in its high-efficiency zone while limiting the state of charge (SOC) variations of the power battery, avoiding overcharging and over-discharging to extend battery life.” (see line 219-221 on page 12)

The specific content is “The input signal needs to be fuzzified before the fuzzy controller is built, and then the fuzzy output is derived from the established fuzzy rule base.” (see line 347-348 on page 19)

2. Could you please revise the introduction so that each paragraph starts with the definition of the keyword it tends to introduce?

Response: Thank you for pointing out the shortcomings, we have revised the paragraph starting with the missing keyword in the introduction and explained the keyword. Please contact us if you have any other suggestions.

The specific content is “The energy management strategy (EMS) of hybrid vehicles is the core technology, which allocates the torque distribution between the engine and the electric motor in real time during the driving process of the whole vehicle, and the goodness of the EMS affects the economy of the whole vehicle to a large extent.” (see line 45-48 on page 3)

The specific content is “The rule-based deterministic energy management strategy assigns fixed control parameters and switches the vehicle to different modes based on its current state, adhering to predetermined limits. This approach primarily relies on subjective experience to optimize economic efficiency.” (see line 54-56 on page 3)

The specific content is “The fuzzy rule-based energy management strategy fuzzyfies the precise values of inputs and uses fuzzy control rules to make decisions on the operating state of the vehicle to achieve regulation and control of the system, thereby improving the adaptability and robustness of energy management.” (see line 65-68 on page 4)

The specific content is “The energy management strategy based on global optimization is based on a cost function that has been defined and a minimum cost function for the driving conditions, resulting in the best fuel economy for the vehicle.” (see line 77-79 on page 4)

The specific content is “Energy management strategies based on transient optimization are divided into equivalent fuel minimum energy management strategies and model predictive energy management strategies. Transient optimization involves selecting the equivalent fuel consumption or total power consumption as the objective function for optimization at each moment of vehicle operation, thus determining the instantaneous optimal operating point for distributing torque between the engine and the electric motor.” (see line 89-94 on page 5)

3. Revise the introduction such that each paragraph starts with the definition of the keyword that it tends to introduce.

Comment: There is a need to restructure the introduction section; concept definition, theoretical review, and empirical review could be easily fetched. Note that each paragraph should announce at least an important study concept. Recall that the major components of a standard paragraph under the introduction are (a) The definition of the term/concept/idea is needed to understand the exact contribution of the report to the body of knowledge. (b) The theoretical review tells us about the published aim and significant theory. (c) The empirical review tells us about published results within the scope of the subject matter.

Response: Thank you for your very constructive suggestions, and as stated in revision 2, we have included definitions of keywords at the beginning of each paragraph of the introduction and explained the concept of keyword learning to make the introduction clearer and more readable.

The specific content is “The rule-based deterministic energy management strategy assigns fixed control parameters and switches the vehicle to different modes based on its current state, adhering to predetermined limits. This approach primarily relies on subjective experience to optimize economic efficiency.” (see line 54-56 on page 3)

The specific content is “The fuzzy rule-based energy management strategy fuzzyfies the precise values of inputs and uses fuzzy control rules to make decisions on the operating state of the vehicle to achieve regulation and control of the system, thereby improving the adaptability and robustness of energy management.” (see line 65-68 on page 4)

The specific content is “The energy management strategy based on global optimization is based on a cost function that has been defined and a minimum cost function for the driving conditions, resulting in the best fuel economy for the vehicle.” (see line 77-79 on page 4)

The specific content is “Energy management strategies based on transient optimization are divided into equivalent fuel minimum energy management strategies and model predictive energy management strategies. Transient optimization involves selecting the equivalent fuel consumption or total power consumption as the objective function for optimization at each moment of vehicle operation, thus determining the instantaneous optimal operating point for distributing torque between the engine and the electric motor.” (see line 89-94 on page 5)

4. The abstract lacks clear separation between methods and results. Could you please separate the description of the methodology from the results for clarity. This helps readers understand the sequence of steps taken in the research.

Response: Thank you for your suggestion, as you said we lacked a clear distinction in our abstract, as you suggested we added a clear analysis of the results in the abstract to make it easier for the reader to understand the steps of the article. If you have any other suggestions, please do not hesitate to give us your feedback!

The specific content is “Compared to traditional rule-based energy management strategies, the optimized fuzzy control using the enhanced dung beetle algorithm continuously adjusts the torque distribution between the engine and motor based on the vehicle's real-time state, resulting in a 9.07% reduction in fuel consumption and a 3.43% decrease in battery SOC fluctuations.” (see line 22-26 on page 2)

5. It was written ::

First, the rule-based energy management strategy is established by dividing the efficient working areas of methanol engine and power battery. Secondly, this paper employs Tent chaotic mapping to integrate the strategies of cosine, Lévy flight, and Cauchy Gaussian mutation, thus improving the dung beetle algorithm. This helps compensate for the traditional dung beetle algorithm's tendency to fall into local optima and enhances the algorithm's global search capability. And the fuzzy controllers for driving charging mode and hybrid driving mode are designed under the rule-based energy management strategy.

Could you revise or consider replacement with:

First, the rule-based energy management strategy is established by dividing the efficient working areas of the methanol engine and power battery. The Tent chaotic mapping is then used to integrate strategies of cosine, Lévy flight, and Cauchy Gaussian mutation, improving the dung beetle algorithm. This integration compensates for the traditional dung beetle algorithm's tendency to fall into local optima and enhances its global search capability. Subsequently, fuzzy controllers for the driving charging mode and hybrid driving mode are designed under this rule-based energy management strategy.

Response: Thank you very much for pointing out our shortcomings, we think you have expressed it more clearly and accurately, and we have taken your suggestion! Thank you for your very constructive comments! (see line 14-20 on page 1)

6. Line 31 - 35, it was written:

Hybrid electric vehicles can optimize energy management and thermal management systems to enhance heat transfer efficiency within the vehicle's energy transmission system. According to Animasaun, I.L et al. [1] the efficiency of heat transfer processes significantly impacts energy shortages by determining the effectiveness of energy utilization and conservation within a system, thereby influencing overall energy demand and supply dynamics.

Comment: Could you please revise and make it more concise? You may consider the replacement below:

Hybrid electric vehicles (HEVs) can optimize energy management and thermal management systems to enhance heat transfer efficiency within the vehicle's energy transmission system. According to Animasaun et al. [1], the efficiency of heat transfer processes plays a crucial role in addressing energy shortages. Efficient heat transfer determines the effectiveness of energy utilization and conservation within a system, directly influencing overall energy demand and supply dynamics.

Response: Thank you very much for your suggestion, we have used your formulation to make the narrative of our article clearer and more accurate! (see line 32-37 on page 2)

7. Line 52, it was written: Jeoung H et al. [8] obtains the optimal logic

Comment: Revise to: Jeoung et al. [8] obtains the optimal logic

Response: Thank you for your suggestion. We have corrected these issues in the article after careful scrutiny. We appreciate your feedback; if you have any further suggestions for changes, please feel free to contact us! (Use blue markings for changes)

8. Line 74, it was written: ... develop control rules. Anselma PG et al.[17] added the slope weighting method in dynamic...

Comment: Revise to:

... develop control rules. Anselma et al.[17] added the slope weighting method in dynamic...

Response: As indicated in Modification Suggestion 7, we have actively revised the deficiencies you pointed out and corrected them throughout the text, thank you for your feedback! (Use blue markings for changes)

9. Validation of the new results seems necessary.

Comment: The discussion of results must not stand in isolation. The authors need to compare the new results in the report with the existing related results in the literature. Consider how the results of the research article align with or differ from previous studies in the field. Discuss any similarities or contradictions and offer possible explanations for the differences. This is the best way to reduce the gap between the introduction and discussion of results.

Response: Thank you for your suggestions, they make our article clearer and we discuss the results in a way that explains the differences between our approach and theirs and compares them with the energy management control strategies used in today's real-world projects. Thank you for your suggestions, and if you have any others, please feel free to contact us!

The specific content is “Unlike most scholars who only add a single fuzzy controller in drive mode, this paper incorporates fuzzy controllers in both linear charging and hybrid drive modes, allowing for a better division of the battery SOC working area in different modes and effectively addressing the issue of excess engine output power. Compared to the currently most widely used rule-based energy management strategy, the improved DBO-optimized multi-fuzzy control in this paper reduces the overall methanol consumption of the vehicle by 9.07% and the fluctuation of the battery SOC by 3.43%, effectively enhancing the vehicle's economy, decreasing the fluctuation of the battery SOC, and greatly improving the adaptability and robustness issues of the rule-based energy management strategy.” (see line 461 on page 25 and see line 462-469 on page 26)

Reviewer: 3(No Response)

We have tried our best to improve the paper and made some changes in the manuscript. These changes will not influence the content and framework of the paper.

We appreciate for Editors/Reviewers’ warm work earnestly, and hope that the correction will meet with approval. We hope that these revisions are satisfactory and will be acceptable for publication in PLOS ONE. Once again, thank you very much for your comments and suggestions.

Wish you all the best.

Sincerely yours,

Zhihao Li, Ping Xiao, Jiabao Pan, Wenjun Pei, Aoning Lv

---

## [Decision Letter · Decision Letter 2]

18 Oct 2024

PONE-D-24-20196R2Energy Management Strategy for Methanol Hybrid Commercial Vehicles Based on Improved Dung Beetle Algorithm OptimizationPLOS ONE

Dear Dr. Xiao,

Thank you for submitting your manuscript to PLOS ONE. After careful consideration, we feel that it has merit but does not fully meet PLOS ONE’s publication criteria as it currently stands. Therefore, we invite you to submit a revised version of the manuscript that addresses the points raised during the review process.

**ACADEMIC EDITOR:**

**In the Abstract section, SOC should be brought with the full words in the first time of using it. In addition, EMS has been defined twice in row 45 and 48, that the second one is redundant. I kindly ask you to carefully check the entire manuscript and correct such issue related to defining and using Acronyms/Abbreviations.**

We look forward to receiving your revised manuscript.

Kind regards,

Mazyar Ghadiri Nejad, Ph.D.

Academic Editor

PLOS ONE

Journal Requirements:

Reviewers' comments:

Reviewer's Responses to Questions

**Comments to the Author**

1. If the authors have adequately addressed your comments raised in a previous round of review and you feel that this manuscript is now acceptable for publication, you may indicate that here to bypass the “Comments to the Author” section, enter your conflict of interest statement in the “Confidential to Editor” section, and submit your "Accept" recommendation.

Reviewer #2: All comments have been addressed

2. Is the manuscript technically sound, and do the data support the conclusions?

Reviewer #2: Yes

3. Has the statistical analysis been performed appropriately and rigorously? 

Reviewer #2: Yes

4. Have the authors made all data underlying the findings in their manuscript fully available?

Reviewer #2: Yes

5. Is the manuscript presented in an intelligible fashion and written in standard English?

Reviewer #2: Yes

6. Review Comments to the Author

**Reviewer #2: **The manuscript reports original scientific research that is of outstanding scientific importance. Based on the content of the latest revised manuscript, it is worth concluding that

a) the manuscript contains an exciting and novel aim,

b) the title is informative and relevant,

c) the introduction, literature review, methodology, results, discussion of results, conclusion, and references are of a high standard,

d) Author(s) have rigorously revised the manuscript. The present form of the whole report is also of a high standard, and

e) the report's contribution to the body of knowledge is significant.

Based on the abovementioned facts, it is worth concluding that the article is error-free and suitable for publication. Therefore, I recommend "Acceptance." Congratulations to the authors for updating the body of knowledge with new scientific facts.

7. PLOS authors have the option to publish the peer review history of their article (what does this mean?). If published, this will include your full peer review and any attached files.

Reviewer #2: No

---

## [Author Response · Author response to Decision Letter 2]

21 Oct 2024

Li, Zhihao

1.Anhui Province Key Laboratory of Intelligent Car Wire-Controlled Chassis System, Anhui Polytechnic University, Wuhu 241000, China

2.School of Mechanical and Automotive Engineering, Anhui Polytechnic University, Wuhu 241000, China

2220110119@stu.ahpu.edu.cn

Xiao, Ping (Corresponding Author)

1.Anhui Province Key Laboratory of Intelligent Car Wire-Controlled Chassis System, Anhui Polytechnic University, Wuhu 241000, China

2.School of Engineering, University of Brigeport, CT 06604, USA

tlxp95@ahpu.edu.cn

13866376423

Pan, Jiabao

1.Anhui Province Key Laboratory of Intelligent Car Wire-Controlled Chassis System, Anhui Polytechnic University, Wuhu 241000, China

2.School of Mechanical and Automotive Engineering, Anhui Polytechnic University, Wuhu 241000, China

panjiabao@ahpu.edu.cn

Pei, Wenjun

1.Anhui Province Key Laboratory of Intelligent Car Wire-Controlled Chassis System, Anhui Polytechnic University, Wuhu 241000, China

2.School of Mechanical and Automotive Engineering, Anhui Polytechnic University, Wuhu 241000, China

2220110151@stu.ahpu.edu.cn

Lv, Aoning

1.School of Mechanical and Automotive Engineering, Anhui Polytechnic University, Wuhu 241000, China

18110850667@163.com

Dear Editors,

Thank you very much for your supervision of the reviewing process of our manuscript with the reference number of PONE-D-24-20196. We also highly appreciate the reviewers’ and editors’ carefulness, conscientiousness, and the broad knowledge on the relevant research fields, since they have given us a number of beneficial suggestions. According to the reviewers’ and editors’ criticism and instruction, we have made the following revisions in terms of format and contents (which are marked in red letters in manuscript):

Academic Editor:

1.In the Abstract section, SOC should be brought with the full words in the first time of using it. In addition, EMS has been defined twice in row 45 and 48, that the second one is redundant. I kindly ask you to carefully check the entire manuscript and correct such issue related to defining and using Acronyms/Abbreviations.

Response: Thank you for pointing out our mistakes, through your suggestions we have revised the manuscript, perfected the problems arising from the acronyms and marked them with red font, if there are any other problems please do not hesitate to contact us. (Use red markings for changes)

2.While revising your submission, please upload your figure files to the Preflight Analysis and Conversion Engine (PACE) digital diagnostic tool, https://pacev2.apexcovantage.com/. PACE helps ensure that figures meet PLOS requirements. To use PACE, you must first register as a user. Registration is free. Then, login and navigate to the UPLOAD tab, where you will find detailed instructions on how to use the tool. If you encounter any issues or have any questions when using PACE, please email PLOS at <a href="mailto:figures@plos.org">figures@plos.org. Please note that Supporting Information files do not need this step.

Response: Thank you for your suggestion, we have used the tool recommended by your journal in order to make the images more in line with the journal standards and re-uploaded the diagrams modified by the tool, we hope that our operation will meet the demand. If there are any other suggestions, please contact us.

Journal Requirements

1.Please review your reference list to ensure that it is complete and correct. If you have cited papers that have been retracted, please include the rationale for doing so in the manuscript text, or remove these references and replace them with relevant current references. Any changes to the reference list should be mentioned in the rebuttal letter that accompanies your revised manuscript. If you need to cite a retracted article, indicate the article’s retracted status in the References list and also include a citation and full reference for the retraction notice.

Response: Thank you for pointing out our problem, we reviewed all references according to the journal's requirements, confirmed that all references were not withdrawn, and added doi numbers! We have also highlighted the revised references in red. If there are any other issues, please feel free to let us know and we would appreciate it. (Use red markings for changes)

We have tried our best to improve the paper and made some changes in the manuscript. These changes will not influence the content and framework of the paper.

We appreciate for Editors/Reviewers’ warm work earnestly, and hope that the correction will meet with approval. We hope that these revisions are satisfactory and will be acceptable for publication in PLOS ONE. Once again, thank you very much for your comments and suggestions.

Wish you all the best.

Sincerely yours,

Zhihao Li, Ping Xiao, Jiabao Pan, Wenjun Pei, Aoning Lv

---

## [Editor Report · Decision Letter 3]

23 Oct 2024

Energy Management Strategy for Methanol Hybrid Commercial Vehicles Based on Improved Dung Beetle Algorithm Optimization

PONE-D-24-20196R3

Dear Dr. Ping Xiao,

We’re pleased to inform you that your manuscript has been judged scientifically suitable for publication and will be formally accepted for publication once it meets all outstanding technical requirements.

Kind regards,

Mazyar Ghadiri Nejad, Ph.D.

Academic Editor

PLOS ONE

---

## [Editor Report · Acceptance letter]

25 Oct 2024

PONE-D-24-20196R3

PLOS ONE

Dear Dr. Xiao,

I'm pleased to inform you that your manuscript has been deemed suitable for publication in PLOS ONE. Congratulations! Your manuscript is now being handed over to our production team.

Kind regards,

on behalf of

Assoc. Prof. Dr. Mazyar Ghadiri Nejad

Academic Editor

PLOS ONE